# The unfolded protein response regulates ER exit sites via SNRPB-dependent RNA splicing and contributes to bone development

Muhammad Zahoor [1,2,10] ✉, Yanchen Dong[3,4,10], Marco Preussner [5], Veronika Reiterer[6], Sabrina Shameen Alam[3,4], Margot Haun[6], Utku Horzum [6], Yannick Frey[6], Renata Hajdu [6], Stephan Geley[6], Valerie Cormier-Daire[7], Florian Heyd [5], Loydie A Jerome-Majewska [3,4,8,9] ✉ & Hesso Farhan [1,6] ✉

## Abstract

Splicing and endoplasmic reticulum (ER)-proteostasis are two key processes that ultimately regulate the functional proteins that are produced by a cell. However, the extent to which these processes interact remains poorly understood. Here, we identify SNRPB and other components of the Sm-ring, as targets of the unfolded protein response and novel regulators of export from the ER. Mechanistically, The Sm-ring regulates the splicing of components of the ER export machinery, including Sec16A, a component of ER exit sites. Loss of function of SNRPB is causally linked to cerebro-costo-mandibular syndrome (CCMS), a genetic disease characterized by bone defects. We show that heterozygous deletion of SNRPB in mice resulted in bone defects reminiscent of CCMS and that knockdown of SNRPB delays the trafficking of type-I collagen. Silencing SNRPB inhibited osteogenesis in vitro, which could be rescued by overexpression of Sec16A. This rescue indicates that the role of SNRPB in osteogenesis is linked to its effects on ER-export. Finally, we show that SNRPB is a target for the unfolded protein response, which supports a mechanistic link between the spliceosome and ER-proteostasis. Our work highlights components of the Sm-ring as a novel node in the proteostasis network, shedding light on CCMS pathophysiology.

**Keywords** COPII; Endoplasmic Reticulum; Sec16A; Proteostasis; Splicing
**Subject Categories** Molecular Biology of Disease; Organelles

## Introduction

Proteostasis refers to the intricate balance between protein synthesis, trafficking, and degradation (Klaips et al, 2018; Plate and Wiseman, 2017). The endoplasmic reticulum (ER) plays a crucial role in proteostasis as it handles one-third of the proteome, making it a major hub for maintaining a balanced proteome. Newly synthesized secretory proteins leave the ER in a COPII-dependent manner at specialized domains referred to as ER exit sites (ERES) (Kurokawa and Nakano, 2019; Peotter et al, 2019). ERES represents an important node in the proteostasis network because they are major determinants of ER unloading. If the function or capacity of ERES does not match the number of proteins queuing for export, then the result is an overload of the ER, which will trigger the unfolded protein response (UPR). The UPR involves three transmembrane proteins, IRE1α, PERK, and ATF6α, which possess stress-sensing domains in the ER and signaling effector domains in the cytosol (Walter and Ron, 2011). The objective of the UPR is to restore ER homeostasis by increasing the folding capacity of the ER by inducing chaperones (Bakunts et al, 2017). In addition, the UPR increases the ER-export capacity by inducing the expression of ERES regulators (Farhan et al, 2008), which underscores the important role of ERES in ER-proteostasis.

Previous comprehensive screenings have focused on identifying regulators of the secretory pathway, leading to the discovery of novel ERES regulators (Farhan et al, 2010; Simpson et al, 2012; Wendler et al, 2010). However, these screens may have overlooked certain regulators, such as components of the splicing machinery, which are largely absent. Alternative splicing has been shown to be involved in ER export regulation (Wilhelmi et al, 2016), but little is known about the role of general splicing in ER proteostasis. Given that the UPR increases mRNA expression of genes encoding ERES

[1]Institute of Basic Medical Sciences, University of Oslo, Oslo, Norway. [2]Centre for Cancer Cell Reprogramming, Institute of Clinical Medicine, University of Oslo, Oslo, Norway. [3]Research Institute of the McGill University Health Centre at Glen Site, Montreal, QC H4A 3J1, Canada. [4]Department of Human Genetics, McGill University, Montreal, QC H3A 0G1, Canada. [5]Institute of Chemistry and Biochemistry, Freie Universität Berlin, Laboratory of RNA Biochemistry, Takustrasse 6, 14195 Berlin, Germany. [6]Institute of Pathophysiology, Medical University of Innsbruck, Innrain 80, 6020 Innsbruck, Austria. [7]Clinical Genetics Department, Université de Paris, INSERM UMR 1163, Imagine Institute, Necker Enfants Malades Hospital, Paris, France. [8]Department of Anatomy and Cell Biology, McGill University, Montreal, QC H3A 2B2, Canada. [9]Department of Pediatrics, McGill University, Montreal, QC H4A 3J1, Canada. [10]These authors contributed equally: Muhammad Zahoor, Yanchen Dong. ✉E-mail: mzkhaans@yahoo.com; loydie.majewska@mcgill.ca; hesso.farhan@i-med.ac.at

components, it is plausible that mRNA splicing contributes to ERES regulation.

To address this knowledge gap, we focused on the spliceosomal subunit SNRPB for two main reasons: firstly, SNRPB was identified in a genome-wide screen for factors regulating autophagy, a process closely linked to ERES structure and function (Cui et al, 2019; Farhan et al, 2017; Zahoor and Farhan, 2018). Secondly, genetic alterations leading to reduced SNRPB expression have been linked to Cerebro-costo-mandibular syndrome (CCMS), a disorder characterized by skeletal malformations (Bacrot et al, 2015; Lynch et al, 2014). However, the specific role of SNRPB in bone development and its pathophysiological basis remains unclear. While we are aware that a defect in splicing might result in bone alterations due to multiple pathways, we think that a defect of ERES function is a strong candidate. Mutations in various components of the COPII machinery or UPR signaling pathway have been associated with bone formation disorders (Boyadjiev et al, 2006; El-Gazzar et al, 2023; Garbes et al, 2015; Horiuchi et al, 2016; Zheng et al, 2021). This commonality led us to hypothesize a potential link between SNRPB and the ER proteostasis network. SNRPB is a Sm protein that together with other Sm proteins (D1-3, E, F, and G) forms the Sm-ring, which is a core scaffold of the U1, U2, U4, and U5 small ribonuclear proteins (snRNPs) (Schwer et al, 2016).

In this work, we show that SNRPB and other components of the Sm-ring regulate the splicing of Sec16A, an ERES regulator. Silencing SNRPB reduced the levels of Sec16A as well as other components of the ER-export machinery. We found SNRPB to be induced by the UPR signal transducer ATF6, and to thereby mediate ERES biogenesis under conditions of secretory cargo overload. Our work thereby establishes an unprecedented role for mRNA splicing in the regulation of ER proteostasis.

# Results

## SNRPB regulate ER-to-Golgi trafficking

We first determined whether loss of SNRPB affects ER-to-Golgi trafficking using the retention-using-selective-hook (RUSH) technique (Boncompain et al, 2012), which monitors the synchronous release of a wave of fluorescently tagged proteins from the ER. We tested two types of RUSH reporters, which were characterized by us and others in the past: Mannosidase-II (Boncompain et al, 2012; Phuyal et al, 2022) and collagen-X (Stadel et al, 2015). Knockdown of SNRPB by two different siRNAs (Appendix Fig. S1A) affected the trafficking of both RUSH reporters (Fig. 1A,B), indicating a possible broad effect on ER export that is not dependent on the type of cargo. Using live imaging mannosidase-II trafficking, we observed that the trafficking defect is a delay in trafficking, rather than a complete block (Appendix Fig. S1B). To account for the fact that the RUSH system relies on overexpression and synchronized trafficking waves, we performed immunofluorescence staining for endogenous ERGIC-53, a cargo receptor that cycles between the ER and the ER–Golgi intermediate compartment (ERGIC). The number of peripheral ERGIC-53 positive puncta is dependent on intact export from the ER, and blocking this trafficking step results in reduced ERGIC puncta (Ben-Tekaya et al, 2005; Farhan et al, 2010). In line with our RUSH experiments, we observed that

silencing SNRPB resulted in a reduction in the number of peripheral ERGIC-53 puncta (Fig. 1C).

Loss of function of SNRPB underlies the rare genetic disease CCMS, which is characterized by bone abnormalities such as micrognathia and rib defects. In light of the effect of SNRPB on ER export, we hypothesized that loss of SNRPB might cause defects in the trafficking of type-I collagen. To investigate the trafficking of endogenous type-I collagen, we depleted SNRPB from primary human fibroblasts, cultured in the absence of ascorbic acid. After 72 h, cells were treated with 500 µg/ml ascorbic acid for 40 min to induce maturation of collagen, followed by fixation and staining for GM130 to label the Golgi apparatus. We measured the amount of type-I collagen in the Golgi region before and after ascorbic acid treatment and noticed that silencing SNRPB resulted in a clear retardation of intracellular collagen trafficking (Fig. 1D). To support this notion, we performed immunofluorescence staining in neural crest tissue from wild type mice and Snrpb neural crest-specific heterozygous ($Snrpb^{ncc+/-}$), which we described recently (Alam et al, 2022). In the tissue of wild-type mice, collagen-I was distributed mostly extracellularly and to some extent intracellularly (Appendix Fig. S2). However, collagen-I exhibited a different staining pattern in tissue from $Snrpb^{ncc+/-}$ mice, with a stronger intracellular distribution. This result is in line with the observed trafficking defect of collagen-I (Fig. 1D).

To gain further insight into the mechanism of the retardation of ER-to-Golgi transport, we determined the number of ERES by staining for Sec31. Silencing SNRPB resulted in a marked decrease of ERES (Fig. 2A). The effect of SNRPB on ERES was also confirmed in the U2OS cell line (Appendix Fig. S1C). Depletion of SNRBP2, which is not part of the Sm-ring, did not affect ERES number (Fig. 2A), which is in line with the absence of an effect on ER-to-Golgi transport. SNRPB is mutated in CCMS patients, and therefore, we examined ERES in fibroblasts from a patient with a mutation on 165 G > C in CCMS (Bacrot et al, 2015). To rescue the expression of this spliceosomal component, we transfected the patient-derived cells with a plasmid encoding GFP-tagged wild-type SNRPB, which localized to the nucleus (Fig. 2B), similar to endogenous SNRPB (Appendix Fig. S3). After 24 h, cells were fixed and immunostained against Sec31 to label ERES. When we compared cells expressing SNRPB with non-transfected cells, we observed a marked increase in the number of ERES (Fig. 2B), which is in line with the role of SNRPB in the regulation of ER export. Altogether, these data show that the spliceosomal subunit SNRPB regulates ER-to-Golgi trafficking by affecting ERES number and function.

## ER-to-Golgi trafficking is regulated by Sm-ring components, but not by other SNRPs

SNRPB is part of the Sm-ring together with SNRPD1-3, SNRPE, SNRPF, and SNRPG. We asked whether the knockdown of components of the Sm-ring affects the levels of the other subunits. Using qPCR, we found that silencing of SNRPB had no major effect on the mRNA levels of the Sm-ring components SNRPD1, SNRPF, and SNRPC (Appendix Fig. S4). SNRPG was found to be upregulated, which might be due to a compensatory effect. We also did not find an effect of SNRPB depletion on the levels of SNRPA and SNRPC (Appendix Fig. S4), which are unrelated SNRPs, and not part of the Sm-ring. Next, we tested whether the

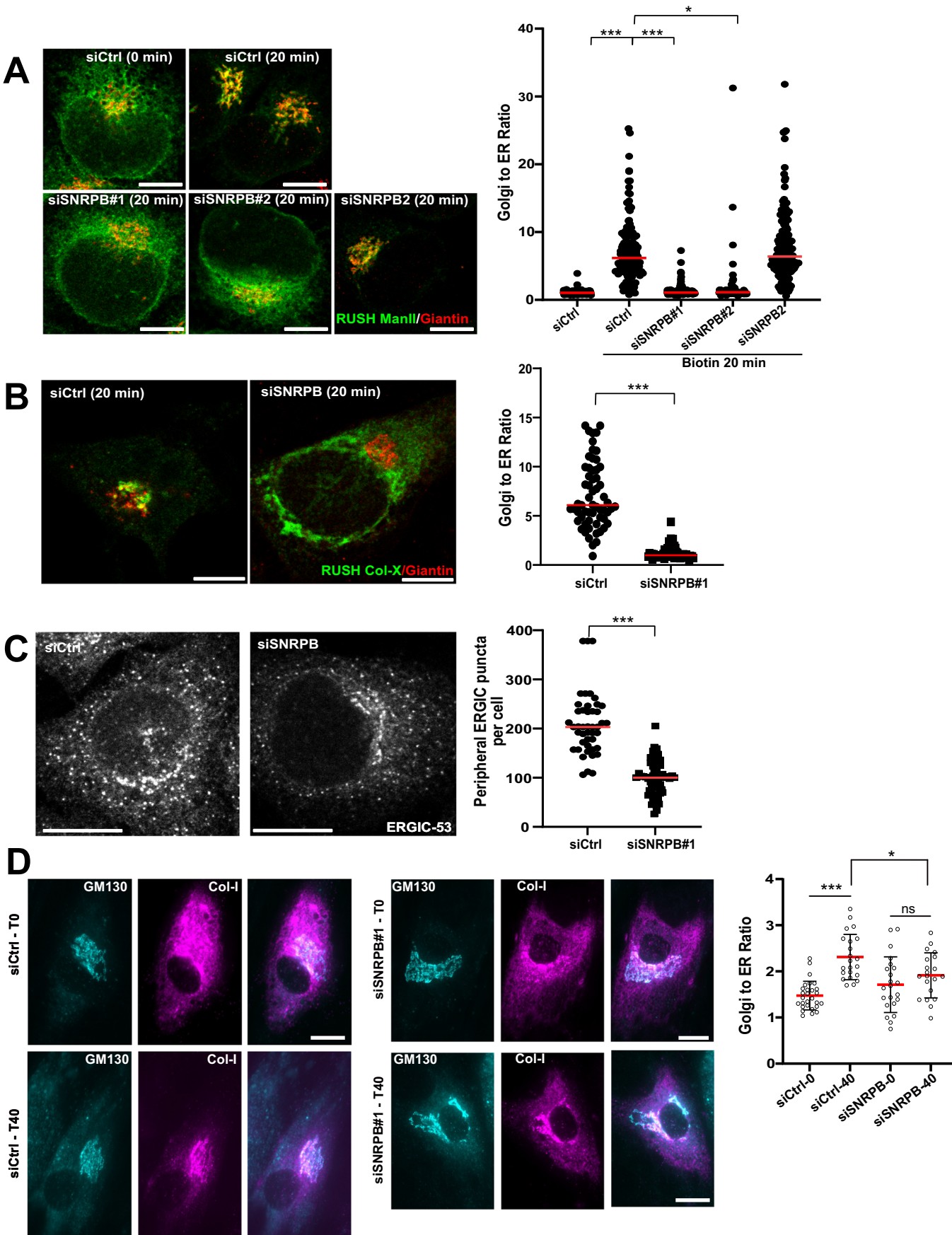

**Figure 1.  Depletion of SNRPB inhibits ER-to-Golgi trafficking.**

(**A**) HeLa cells stably expressing mCherry tagged mannosidase-II RUSH construct (ManII) were transfected with siRNA against SNRPB (siSNRPB) or with a non-targeting control siRNA (siCtrl). After 72 h, cells were treated with biotin (40 μM) for 20 min followed by fixation and immunofluorescence staining against giantin to label the Golgi. The graph to the right shows a quantification of three independent experiments. The fluorescence intensity of the mCherry signal in the Golgi area was normalized to that in the whole cell giving the Golgi-to-ER ratio. *** indicates a *P* value smaller than 0.001 and * indicates a *P* value smaller than 0.05, obtained by one-way ANOVA. Scale bar = 10 μm. (**B**) HeLa cells were transfected with siRNA against SNRPB (siSNRPB) or with a non-targeting control siRNA (siCtrl). After 48 h, cells were transfected with a plasmid encoding the GFP-tagged type-X collagen as part of the RUSH system. After 24 h, cells were treated with biotin (40 μM) for 20 min, followed by fixation and immunofluorescence staining against giantin to label the Golgi. *** indicates a *P* value smaller than 0.001 obtained using a non-paired, two-tailed *t* test. Data are from three independent experiments. Scale bar = 10 μm. (**C**) HeLa cells were transfected with siRNA against SNRPB (siSNRPB) or with a non-targeting control siRNA (siCtrl). After 48 h, cells were fixed and immunostained against ERGIC-53. The graph to the right shows a quantification of three independent experiments where the number of ERGIC puncta was counted. *** indicates a *P* value smaller than 0.01 obtained using a non-paired, two-tailed *t* test. Scale bar = 10 μm. (**D**) Primary human fibroblasts were cultured in the absence of ascorbic acid and transfected with non-targeting siRNA (ciCtrl) or siRNA against SNRPB. After 72 h, cells were fixed (T0) or treated with ascorbic acid for 40 min followed by fixation (T40). Cells were stained for type-I collagen and for GM130 to label the Golgi apparatus. The plot at the right side shows the quantification from three independent experiments with one-way ANOVA. *** indicates a *P* value smaller than 0.001 and * indicates a *P* values smaller than 0.05. Scale bar = 10 μm. Source data are available online for this figure.

effect on ER-to-Golgi trafficking is observed specifically with SNRPB knockdown, or whether any alteration of the Sm-ring would affect secretory trafficking. We therefore silenced the Sm-ring components SNRPB, SNRPD1, and SNRPG and observed that all of them negatively affected ER-to-Golgi trafficking as assayed using a RUSH assay with mannosidase-II as a reporter (Fig. 3A,B). Silencing of the non-related SNRPB2, and SNRPC had no appreciable effect on ER-to-Golgi trafficking (Figs. 3B and 1A). These results reinforce the conclusion of a link between the general spliceosome and ERES function.

## SNRPB knockdown decreases the level of ERES components

We next wanted to explore the cause for the alteration in ERES number and function. To this end, we determined the level of several components of ERES in SNRPB knockdown cells. We noticed a downregulation of several proteins involved in ER-to-Golgi trafficking such as Sec16A, Sar1A, Sec12, Sec24C, and Sec31 (Fig. 4A). There was a weak reduction in the protein levels of Sec13, which we consider to rather be a consequence of the reduction of Sec31A levels, the dimerization partner of Sec13. Two pools exist for Sec13, one that is part of the COPII coat and another one that is part of the nuclear pore complex (Enninga et al, 2003). We think that the weak reduction of Sec13 is because only the COPII-associated pool is affected because of a loss of its dimeric partner Sec31. We also probed for the Golgi matrix protein Giantin, but did not find a difference as was the case for the ER chaperone calnexin (Fig. 4A). We used actin as a loading control, which was also not affected by the depletion of SNRPB.

The reduction of the levels of ERES proteins was not due to a transcriptional effect because the mRNA levels of these proteins did not drop upon SNRPB depletion, but rather went up (e.g., Sar1A) (Fig. 4B). The effect is also unlikely to be a consequence of the alteration of protein stability, because treatment of cells with bortezomib for 4 h (an inhibitor of the proteasome) did not restore the levels of Sec24C, in contrast to β-catenin, which is known to be turned over by proteasomal degradation (Fig. 4C).

## SNRPB mediates the splicing of Sec16A

Because SNRPB is a spliceosomal component, we hypothesized that the downregulation of the levels of ERES proteins is due to

defective splicing of their pre-mRNA. Among all affected proteins, we decided to focus on Sec16A, because it is a general regulator of ERES, that acts as an upstream regulator of all COPII components (Tang, 2017). To investigate a potential splicing defect of Sec16A (e.g., increased intron retention), we designed exon-based primers and amplified a region of the Sec16A and actin transcripts. For technical reasons, we focused on exons 6–8 of Sec16A and designed primers that would give rise to various amplicons due to intron retention or exon skipping (schematically depicted in Fig. 5A). Silencing SNRPB resulted in a marked change in the pattern of bands for Sec16A and an appearance of PCR amplicons indicative of exon skipping or intron retention (Fig. 5A). No effect was observed on actin (Fig. 5A), which is in line with the lack of an effect on SNRPB depletion on the protein levels of actin (Fig. 4A). Silencing SNRPC, which is not part of the Sm-ring, produced a pattern similar to control siRNA transfected cells, which is in line with the absence of an effect of SNRPC knockdown on ER-to-Golgi trafficking (Fig. 3). Silencing of the Sm-ring component SNRPD1 produced a similar effect as SNRPB depletion, indicating that interference with components of the Sm-ring produces effects on Sec16A splicing and ER-to-Golgi trafficking. Similar results were obtained when looking at a different region of the Sec16A mRNA (exons 17–19) where we found that silencing Sm-ring components SNRPB and SNRPD1 to produce a splicing defect, but not silencing of SNRPC (Appendix Fig. S5A).

We reasoned that a splicing defect would result in a retention of the major fraction of Sec16A pre-mRNA in the nucleus. Therefore, we isolated RNA from nuclear and cytosolic fractions and determined Sec16A mRNA levels. The levels of Sec16A mRNA in SNRPB-depleted cells were lower in the cytosolic fraction, and higher in the nuclear fraction (Fig. 5B,C), which is in line with a splicing defect. Nuclear retention of pre-mRNA only occurs when it is associated with the spliceosome (Huang and Carmichael, 1996) which indicates that the knockdown of SNRPB does not totally abolish the formation of the spliceosome, but alters its function.

To better link the effect of SNRPB on ER-export to its effect on Sec16A levels, we performed a rescue experiment using a GFP-tagged Sec16A construct, which we and others have demonstrated to be functional (Farhan et al, 2010; Tillmann et al, 2015). We overexpressed GFP-Sec16A in SNRPB-depleted cells, and monitored ER-to-Golgi trafficking using the RUSH assay. As a control, we overexpressed GFP. Cells overexpressing GFP-Sec16A exhibited a partial restoration of ER-to-Golgi trafficking in SNRPB-depleted

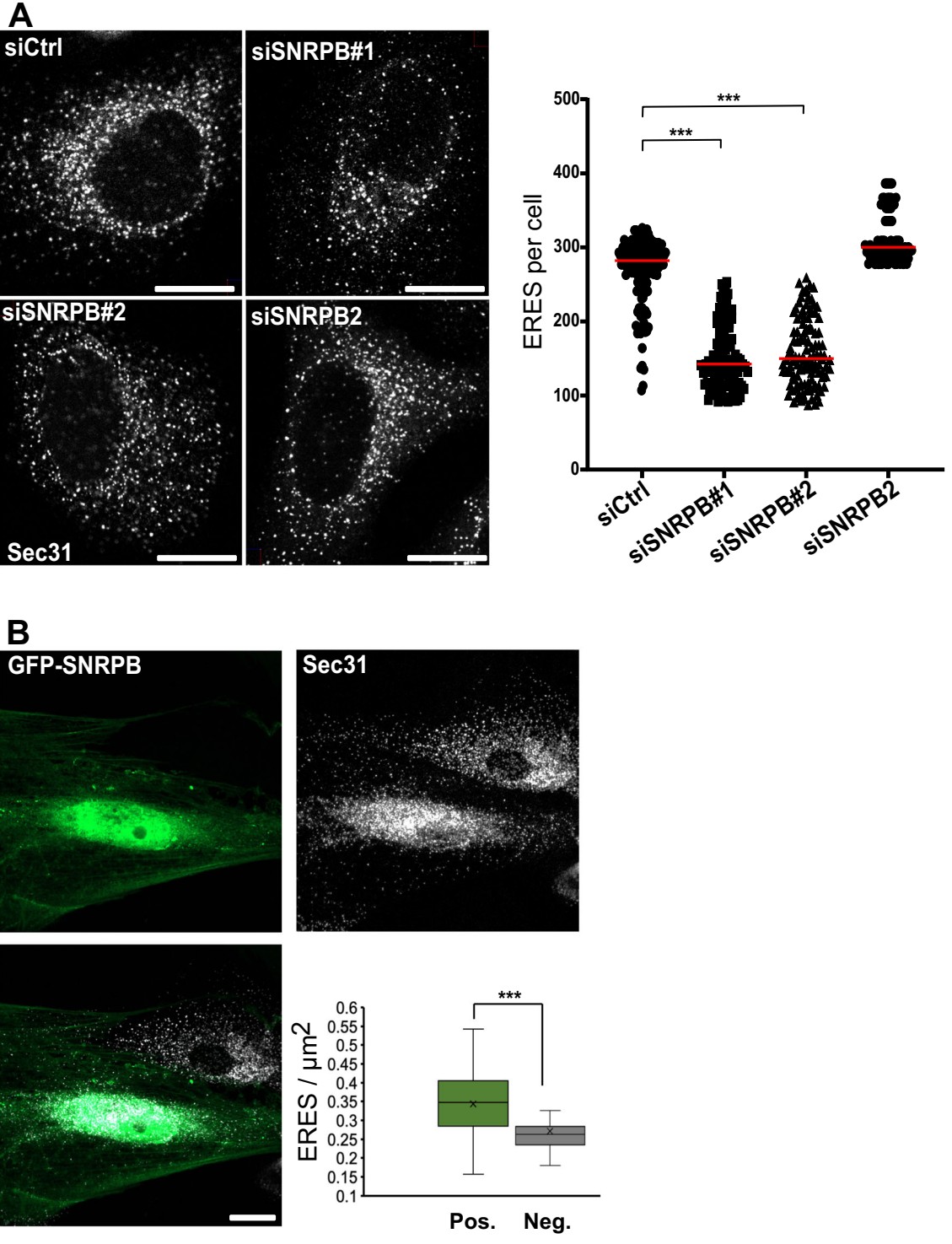

**Figure 2.  SNRPB regulates the number of ERES.**

(**A**) HeLa cells were transfected with siRNA against SNRPB or SNRPB2 or with a non-targeting control siRNA (siCtrl). After 72 h, cells were fixed and immunostained against Sec31 to label ERES. The number of ERES per cell (from three independent experiments) was counted using ImageJ and is displayed in the graph to the right. *** indicates a $P$ value smaller than 0.001 obtained using one-way ANOVA. Scale bar = 10 μm. (**B**) Fibroblasts derived from a patient with SNRPB mutation were electroporated with a plasmid cDNA encoding GFP-tagged SNRPB. After 24 h, cells were fixed and immunostained against Sec31 to label ERES. Box blot shows the quantification of a total of 60 cells from three independent experiments. Pos cells positive for GFP, Neg cells negative for GFP. *** indicates a $P$ value smaller than 0.001 obtained using a non-paired, two-tailed $t$ test. Data were represented as box plots. The interquartile range is presented by a rectangular box with the median as the center of the box. Whiskers indicate the min. to max. range. Scale bar = 10 μm. Source data are available online for this figure.

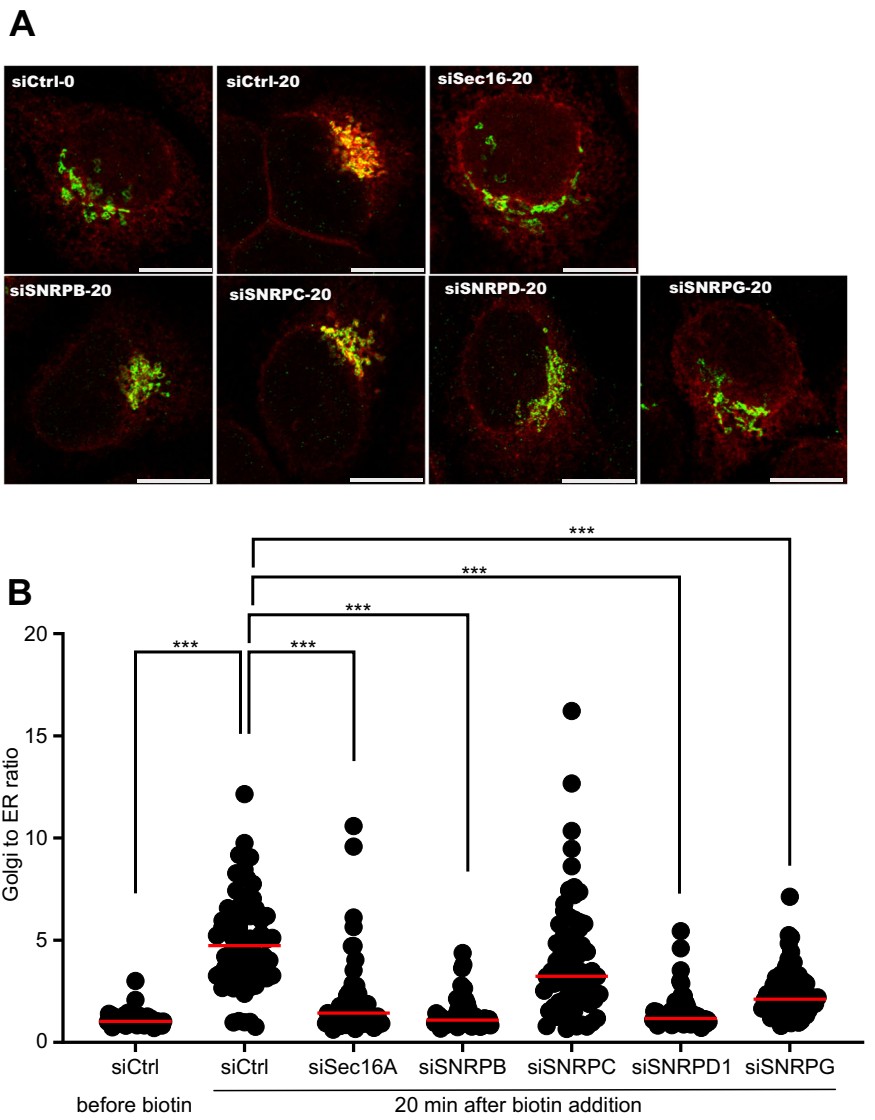

**Figure 3.   Sm ring components regulate ER-to-Golgi trafficking.**

(A, B) HeLa cells expressing mannosidase-II RUSH construct were transfected with siRNA as indicated. siCtrl indicates transfection with non-targeting control siRNA. After 72 h, the RUSH assay was started by treatment with 40 μm biotin followed by fixation and immunostaining for GM130 to label the Golgi. (A) Shows representative merged images and the graph shows a quantification of three independent experiments. The fluorescence intensity of the signal in the Golgi area was normalized to that in the whole cell giving the Golgi-to-ER ratio. *** indicates a *P* value smaller than 0.001 obtained using one-way ANOVA. Scale bar = 10 μm. Source data are available online for this figure.

cells (Fig. 5D). These results indicate that the reduction in the levels of Sec16A in SNRPB silenced cells is the main factor that contributes to the defect in ER export.

## Global transcriptomic analysis reveals an effect of SNRPB on splicing of ERES regulators

To gain a systematic overview of the extent of the effects of SNRPB on the secretory pathway, we performed deep RNA sequencing of cells depleted of SNRPB. Samples were correctly assigned and exhibited strong silencing of SNRPB (Appendix Fig. S6). First, we used Salmon and Deseq2 as an analysis pipeline for differential gene expression and the results are displayed as a volcano plot with

genes related to secretory trafficking highlighted in green (Fig. 6A; Dataset EV1). In line with our qPCR data on Sec16A and other components of the secretory pathway, we did not detect a change in mRNA levels of these genes. Next, we analyzed differential splicing using two pipelines: STAR/RMATS (Fig. 6B; Dataset EV2) or the Whippet (Fig. 6C; Dataset EV3). Both show that SNRPB knock-down results in massive cassette exon skipping (SE) as well as to a lesser extent intron retention (RI) or alternative splice site usage (A5SS or A3SS), which is consistent with previous work (Saltzman et al, 2011). We plotted the RMATS-derived change in exon inclusion in a volcano plot in Fig. 6D, again highlighting exons in ER–Golgi trafficking genes in green. Overall, we identified 40 genes that exhibited affected transcripts. Expanding on these global

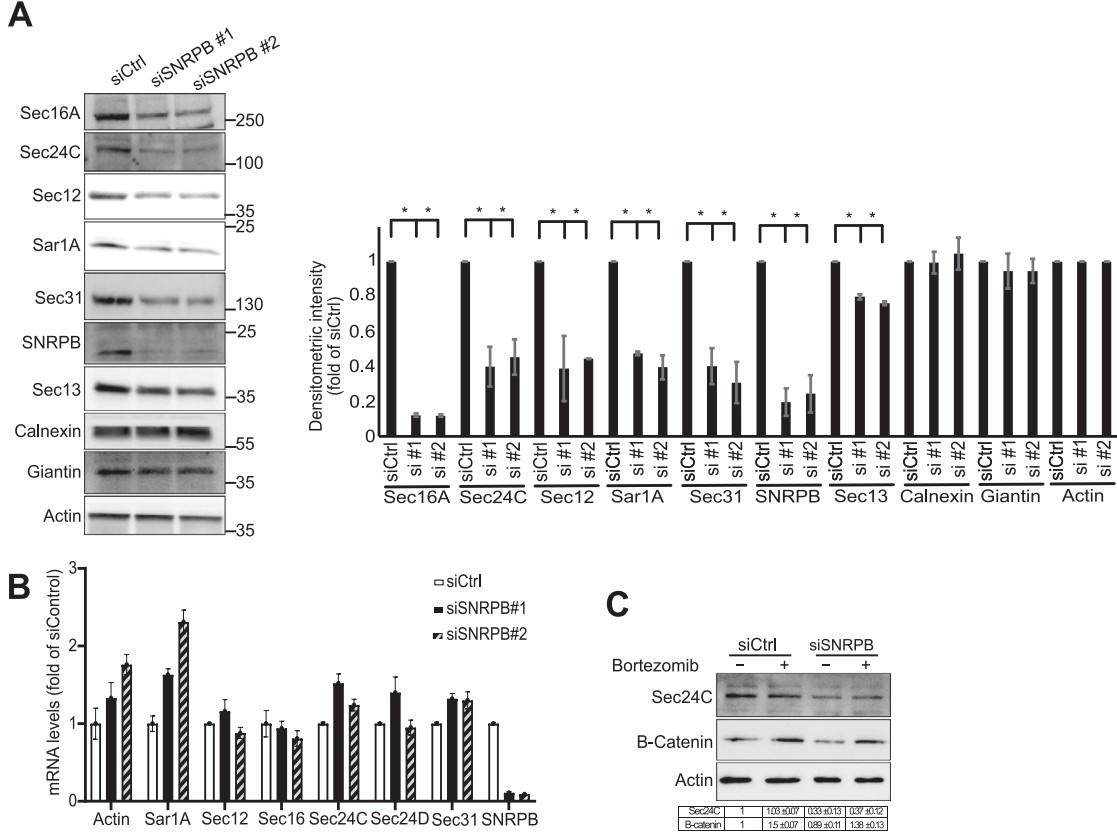

**Figure 4. Depletion of SNRPB regulates the protein levels of ERES components.**

(**A**) HeLa cells were transfected with two siRNAs against SNRPB or with a non-targeting control siRNA (siCtrl). After 72 h, cells were lysed followed by immunoblotting as indicated. The graph on the right side shows the quantification of three independent experiments (±SD). Asterisks indicate statistically significant differences with $P \leq 0.05$ (one-way ANOVA). (**B**) quantitative RT-PCR of the indicated transcripts in HeLa cells depleted of SNRPB using two siRNAs. Data are from 3 experiments (±SD). (**C**) HeLa cells were transfected with siRNA against SNRPB or with a non-targeting control siRNA (siCtrl). After 72 h, cells were treated with bortezomib (+) or solvent (−) followed by lysis and immunoblotting as indicated. The table below the blots shows a quantification of two independent experiments. Source data are available online for this figure.

analyses, manual screening of secretory genes revealed exon skipping and intron retention within a Sec16A region with low sequence coverage (Fig. 6E), therefore escaping our global analysis. Overall, these data indicate that SNRPB regulates the splicing of several secretory pathway genes and that Sec16A is one of the clients of this spliceosome that explains its effect on ERES and ER-to-Golgi trafficking. Among the affected transcripts were other components of the secretory pathway such as GOSR1 (sashimi plot in Appendix Fig. S7B), where exon 7 was affected (Dataset EV2). This was validated by PCR (Appendix Fig. S5B). Accordingly, we also found that SNRPB depletion results in a reduction in protein levels of GOSR1 similar to various ERES components (Appendix Fig. S7A). ERES proteins and GOSR1 were not affected by SNRPB2 depletion (Appendix Fig. S7A). Further PCR validations of splicing defects were performed for exon 20 in Sec31 and exon 2 in ERGIC2 (Appendix Fig. S5C), where are transcripts shown to be affected in our RNAseq analysis.

To determine whether there are common features in affected exons, we extracted five features from exons unregulated ($n = 108147$), skipped ($n = 2804$) or included ($n = 156$) upon SNRPB knockdown. These features were upstream-intron, exon and downstream intron length, as well as upstream 3'ss and

downstream 5'ss score. Exons skipped upon knockdown are slightly shorter and are surrounded by slightly shorter introns. In addition, they are rather characterized by weak 5'ss than differences in 3'ss strength (Appendix Fig. S8).

## SNRPB is regulated by cargo load in a manner dependent on ATF6

Our results so far suggest that SNRPB might regulate the ER-proteostasis network by modulating the function of ERES. We next wanted to gain a broader understanding of how SNRPB regulates ER-proteostasis, and whether SNRPB itself might be regulated by signaling pathways within the proteostasis network. The UPR is a transcriptional response that regulates ER export by inducing the expression of multiple ERES components. We hypothesized that the spliceosomal subunit SNRPB might be involved in mediating the effects of the UPR on ERES. To induce UPR, we decided to use ER-overload, because we found in a previous study that the levels of Sec16A are elevated under conditions of ER-overload, in a manner dependent on the UPR (Farhan et al, 2008). To induce UPR by ER-overload, we overexpressed two constructs: GABA transporter 1 (GAT1) and GFP-b(5)tail. GAT1 is a polytopic transmembrane

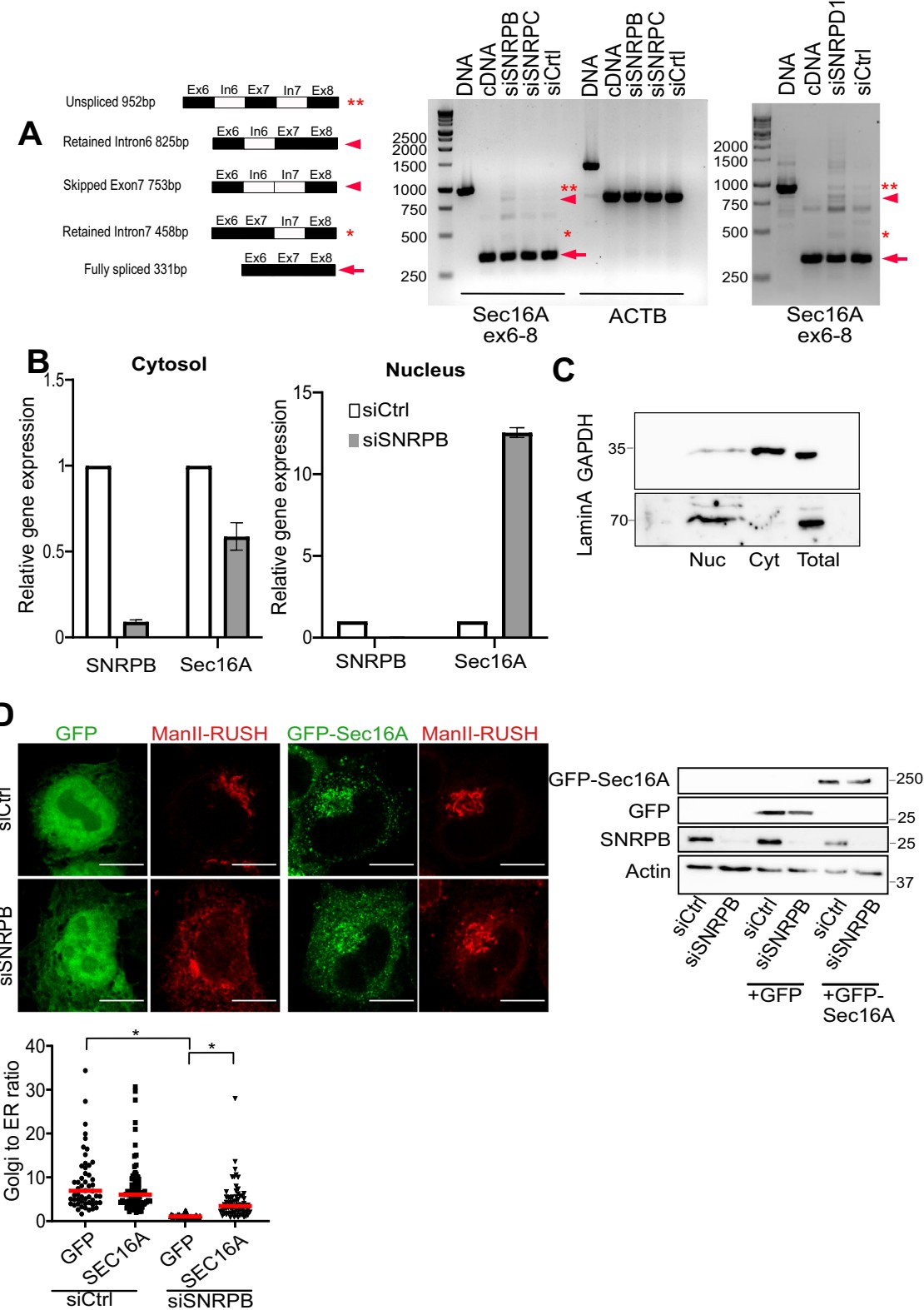

**Figure 5.  Depletion of SNRPB regulates Sec16A splicing.**

(A) The schematic to the left shows the sizes of all possible splicing defects for the region spanning exon 6–8 of Sec16A. These are labeled with symbols, which are found on the scans of the agarose gels. RT-PCR using exon-based primers spanning exons 6–8 of Sec16A and exons 3–5 of beta Actin (ACTB) was performed in cells transfected with the indicated siRNAs. The gel was loaded with DNA, cDNA and the PCR products from the different knockdown conditions. (B) Quantitative real-time RT-PCR for SNRPB and Sec16A of cytosolic and nuclear fractions from cells transfected with non-targeting (siCtrl) or SNRPB siRNA or 72 h. Data are ±SD from three independent experiments. (C) Cells were fractionated as for panel B and lysates were subjected to immunoblotting to test for the purity of the nuclear (Lamin A as a marker) and cytosolic (GAPDH as a marker) fraction. (D) HeLa cells stably expressing mCherry tagged mannosidase-II RUSH construct (ManII) were transfected with siRNA against SNRPB (siSNRPB) or with a non-targeting control siRNA (siCtrl). After 48 h, cells were transfected with plasmids encoding GFP or GFP-tagged Sec16A. After 24 h, cells were treated with biotin for 20 min followed by fixation. The amount of mCherry signal in the Golgi area was normalized to the total cellular mCherry signal. The immunoblot demonstrates the efficiency of SNRPB knockdown and that GFP-Sec16A and GFP were faithfully expressed. The result of three independent experiments with 30 cells per experiment is displayed in the graph to the right. * indicates a P value smaller than 0.05 obtained using one-way ANOVA. Source data are available online for this figure.

protein, which we showed previously to induce the UPR when overexpressed (Farhan et al, 2008). GFP-b(5)tail is a construct composed of the N-terminal cytosolic GFP moiety anchored to the ER membrane by the tail of cytochrome b5, which was shown previously to activate the ATF6 branch of the UPR (Maiuolo et al, 2011). We opted to induce UPR by overexpressing GAT1 and GFP-b(5)tail, instead of using chemical tools such as thapsigargin or tunicamycin, which do not accurately mimic the response to ER overload (Bergmann et al, 2018).

Overexpression of GAT1 as well as GFP-b(5)tail resulted in an induction of Sec16A levels, which was abrogated upon knockdown of SNRPB (Fig. 7A; Appendix Fig. S9A). Furthermore, ER-overload resulted in an increase in the number of ERES, which was absent in SNRPB-depleted cells (Fig. 7B). Similar results were obtained with GFP-b(5)tail overexpression, which increased the levels of Sec16A and the number of ERES in a manner dependent on SNRPB (Fig. 7A,C).

We also noted that overloading the ER also resulted in a clear increase in SNRPB levels (Fig. 7A), suggesting that SNRPB itself is regulated by the UPR. This result would be in line with our hypothesis of SNRPB as a mediator or effector of the UPR. To test this hypothesis, we overexpressed GAT1 in cells depleted of XBP1, ATF4, and ATF6, which are the main signaling branches of the UPR. GAT1 overexpression resulted in an increase in the protein levels of SNRPB in all conditions except in ATF6 knockdown cells (Fig. 7D; Appendix Fig. S9B). The induction of SNRPB was also visible at the mRNA level, which is in agreement with the UPR being a transcriptional response. Again, the induction of SNRPB by ER-overload was abrogated in ATF6-depleted cells (Fig. 7E). Silencing of XBP1 or ATF4 had only a very weak effect indicating that ATF6 is the major pathway that is used to induce SNRPB. We also found that GFP-b(5)tail overexpression induces the mRNA levels of SNRPB, which was abrogated by silencing ATF6 (Fig. 7F). To support the notion of a role of ATF6 in the regulation of SNRPB levels, we silenced ERp18, which is known to regulate ATF6 dimerization and activation (Oka et al, 2022). To test for ATF6 activation, we performed immunofluorescence and assessed its localization to the Golgi apparatus upon treatment with thapsi-gargin. Silencing ERp18 reduced the localization of ATF6 in the Golgi region, indicating that it interferes with the activation of this UPR branch (Appendix Fig. S10). Next, we transfected HeLa cells with GAT1 and observed that the increase in the levels of SNRPB were abrogated by two different siRNAs against ERp18, which supports the notion that induction of SNRPB is (at least partially) controlled by ATF6.

Overall, these results indicate that SNRPB is an effector of the UPR, while at the same time being a facilitator of the effects of the UPR on ER proteostasis (Fig. 7G).

## Loss of SNRPB induces osteogenesis defects in vitro that are rescued by Sec16A overexpression

Loss of function of SNRPB underlies the rare genetic disease CCMS, which is characterized by severe bone deformations, microcephaly, rib defects and, micrognathia (Bacrot et al, 2015; Leroy et al, 1981; McNicholl et al, 1970). We are fully aware that bone development is a complex process that is regulated by multiple factors. Nevertheless, we asked whether the effect of SNRPB on ER-proteostasis might contribute partially to the pathophysiology of CCMS. Previous research linked multiple proteins of the ER-export machinery with bone defects (Boyadjiev et al, 2006; El-Gazzar et al, 2023; Garbes et al, 2015), which supported the notion that the effects of SNRPB on ER-export might be linked to bone defects in CCMS. We, therefore, tested whether loss of SNRPB affects osteogenesis using an in vitro assay where we differentiated mesenchymal cells to osteoblasts. Osteogenesis can be visualized using alizarin red staining, which detects calcification. Silencing SNRPB with two siRNAs resulted in a notable reduction in osteogenesis, as indicated by decreased alizarin red staining intensity (Fig. 8A–B).

To link the effect of SNRPB on osteogenesis to ERES function, we asked whether overexpression of Sec16A might rescue the effect of SNRPB depletion. We chose Sec16A because it plays a key role in the biogenesis and maintenance of ERES, upstream of COPII subunits. In cells with reduced levels of COPII components, overexpression of Sec16A would potentiate the recruitment of the available COPII subunits to ERES to drive ER export. In our previous work, we showed that signaling to Sec16A is sufficient to drive the biogenesis of new ERES without the need to regulate the levels of any COPII component (Tillmann et al, 2015). As shown in Fig. 8A, overexpression of Sec16A almost completely rescued the osteogenesis defect in SNRPB-depleted cells. These results indicate that the bone defects observed under loss of SNRPB are, at least partially, linked to a defect in the ERES function.

## Inducible Snrpb deletion results in chondrogenesis and osteogenesis defects

We previously showed that embryos with heterozygous loss of function mutation in Snrpb arrested at embryonic day (E) 8.5

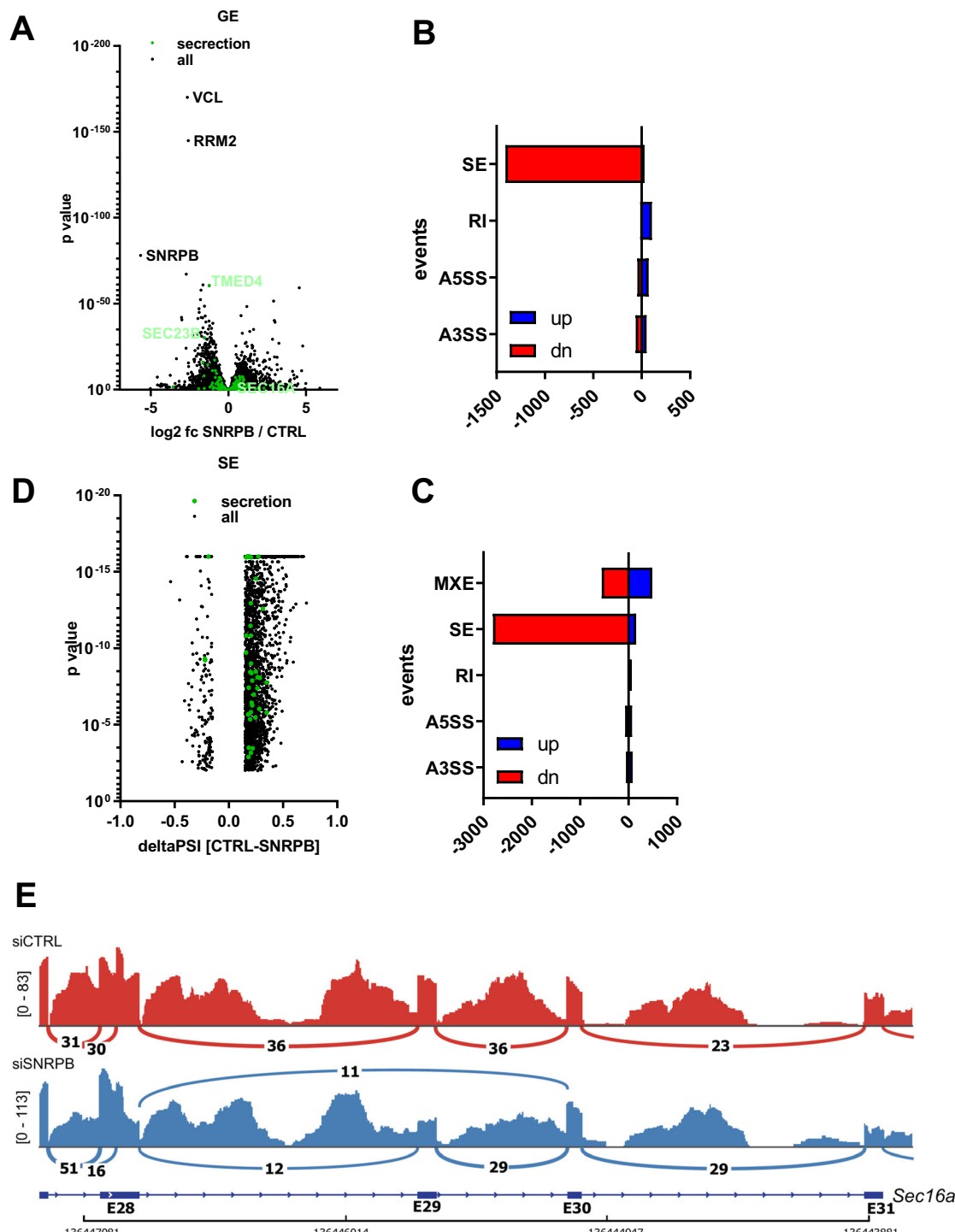

(Alam et al, 2022), indicating an early requirement for this gene during gastrulation. We also showed bone deformities in the craniofacial region of neural crest cell *Snrpb* mutants. To examine the requirement of *Snrpb* in bone development in the appendages, we used tamoxifen to induce the deletion of Snrpb at E8.5. Mice carrying LoxP sequences flanking exon 2 and 3 of *Snrpb* (Snrpb^loxP/+) (23) and ER-Cre^tg/tg (Ventura et al, 2007) were crossed to produce *Snrpb*^loxP/+; *ER-Cre*^g/tg males for mating with wild type

CD1/CD57 females. Pregnant females were injected with a single dose of tamoxifen at E8.5 to induce the deletion of exons 2 and 3 of *Snrpb*. The resulting litters were collected at E14.5 (*n* = 18 controls and 14 mutants) and E17.5 (*n* = 15 controls and 12 mutants) and stained with Alcian blue and Alizarin red to visualize the cartilage and bones, respectively. Below, we describe severe malformations in both limbs and girdles (scapula and pelvis) of *Snrpb*^LoxP/+;*ER-Cre*^Tg/+ embryos at E14.5 and E17.5 (Fig. 9A–F). The scapula was smaller

**Figure 6. Systematic exploration of the effects of SNRPB depletion.**

(A) Volcano plot highlighting SNRPB depletion-induced changes in global gene expression (GE). The normalized fold change in gene expression is plotted on the *x* axis, the Deseq2-derived statistical significance is plotted on the *y* axis (with most significant genes at the top). Genes associated with the GO terms "COPII-coated vesicle budding" and "regulation of regulated secretory pathway" are highlighted in green. See also Dataset EV1. *P* values were generated using Deseq2. (B, C) SNRPB depletion induced changes in alternative splicing derived by RMATS (B) or Whippet (C). Targets with a higher percent splice index (PSI) upon knockdown are highlighted in blue, with a lower PSI in red. MXE mutually exclusive exons, SE skipped exons, RI retained introns, A3SS alternative 3'splice-sites, A5SS alternative 5' splice-sites. Note that both splicing tools reveal massive exon skipping in response to SNRPB depletion. (D) Volcano plot highlighting SNRPB depletion-induced exon skipping. DeltaPSI (*x* axis) and *P* values are based on RMATS. Exons in secretion-associated genes are highlighted as in (A). *P* values were generated using RMATS. (E) Sashimi plot highlighting SNRPB depletion-induced changes in Sec16 splicing. Summarized are all reads from siCTRL (red) or siSNRPB (blue). Pilled sequencing coverage is indicated on the y-axis (number of reads per position). Junction reads are shown as lines (minimum junction reads per junction shown is set to 10). Below the genomic region encoding Sec16 as well as the exon intron structure is indicated. Note E29 skipping and increased IR of the surrounding introns upon depletion of SNRNPB.

and narrower in most mutant embryos ($n = 19/26$) and showed dysmorphologies including a missing blade ($n = 7/26$), bifurcated blade ($n = 7/26$), and fused shoulder joints ($n = 10/26$). In a small subset of E17.5 embryos, a curvy clavicle ($n = 4/12$) was also found. In the pelvis, the iliac bone was thinner ($n = 18/26$) or missing ($n = 9/26$) in *Snrpb* mutant embryos. Defects were also found in the stylopod. The humerus of *Snrpb* mutants was narrower in E14.5 embryos ($n = 3/14$) than those of controls and few were identified with missing the deltoid tuberosity ($n = 2/14$). At E17.5, the deltoid tuberosity was misshapen or smaller in six of the twelve mutant embryos analyzed. In addition, the olecranon process was malformed in mutants of both stages ($n = 4/26$). The femur of *Snrpb* mutants was also thinner and had an abnormal narrowing at the middle ($n = 8/26$). The zeugopod of mutants was often bent, which was more evident in the ulna ($n = 19/26$) when compared to the radius ($n = 17/26$). In addition, fewer embryos had bending in the tibia and fibula ($n = 6/26$). Defects, in the autopod included fused 2nd and 3rd metacarpal ($n = 1/26$), bilaterally, and a missing 5th digit ($n = 1/26$). However, these malformations were rare, with each found in one Snrpb mutant embryo. Moreover, clinodactyly, or bent 5th phalanges was found in both controls ($n = 4/33$) and mutant ($n = 4/26$) embryos. In addition to these dysmorphologies, at E17.5, Alizarin red staining was missing in the ulna and radius of six mutant and four control embryos, or split into two domains in the ulna of one mutant embryo, indicating abnormal osteogenesis. Finally, the humerus, radius, ulna, and femur of mutants ($n = 12$) were shorter when compared to those of controls ($n = 15$) (unpaired, two-tailed *t* test, $P < 0.05$) (Fig. 9G). Hence, reducing levels of *Snrpb* after a single injection of tamoxifen at E8.5, leads to malformations in the appendage of the majority of *Snrpb*[LoxP/+];ER-Cre[Tg/+] embryos ($n = 25/26$), except for one E14.5 mutant embryo. Furthermore, in most E14.5 ($n = 10/14$) and all E17.5 ($n = 12$) mutant embryos these malformations were bilateral, though the severity differed between the left and right side. Altogether our data indicate that normal level of Snrpb is required for growth and patterning of the proximal and distal elements of the mouse appendage, and for both chondrogenesis and osteogenesis.

## Discussion

Research over the past few decades has expanded our understanding of the role of the UPR in regulating cellular proteostasis under physiological and pathological conditions such as cancer and neurodegenerative disorders (Hetz and Saxena, 2017; Urra et al, 2016). The UPR is a transcriptional response and so far, it has been

assumed that no regulatory step exists between the UPR-mediated induction of mRNAs and their translation into proteins. The splicing machinery was never assumed to be a rate-limiting factor in the process of converting UPR-induced mRNAs into proteins. Our current work indicates that this assumption is not correct and expands our understanding of the regulation of ER proteostasis. We show that the spliceosomal component SNRPB is required for the adaptation of the ER to higher loads of secretory cargo where it acts as part of a feed-forward loop that signals from UPR to ERES in an ATF6-dependent manner. In the absence of SNRPB, cargo load failed to induce an increase in the levels of the ER-export machinery. We hypothesize that the induction of several hundred mRNAs by the UPR exceeds the capacity of the pre-existing pool of SNRPB, thus necessitating the induction of this spliceosomal component by ATF6.

Mechanistically, the effect of SNRPB on ER proteostasis was mediated through splicing of components of the membrane trafficking machinery such as regulators of biogenesis and trafficking at ERES. Consequently, silencing of SNRPB reduced the number of ERES and delayed trafficking from the ER to the Golgi. Although several ERES components were affected, we were able to rescue the ERES defect by exogenous expression of Sec16A, indicating that this ERES regulator might represent the dominant component behind the effect of SNRPB on ER proteostasis. In the absence of SNRPB, cells failed to upregulate the levels of Sec16A in response to ER overload. Thus, we propose SNRPB as a new checkpoint in the ER-proteostasis network. We noted that the regulatory circuit has features of a coherent feed-forward loop (CFFL) (Fig. 7G). The biological function of many CFFLs is to act as persistence detectors to ensure that only a persistent stimulus exerts a full biological response (Lim et al, 2013). A feature of CFFLs is that one of the two forward loops is slower than the other. In our case, the loop that includes SNRPB (green in Fig. 7G) contributes in a slower manner towards the output than the loop where the UPR stimulates the transcription of ERES components (purple in Fig. 7G). We hypothesize that the initial UPR response is handled by the pre-existing pool of SNRPB. However, a persistent UPR creates more clients for the snRNPs/spliceosome and therefore necessitates higher levels of SNRPB, and maybe other Sm-ring component. The observation that SNRPB knockdown cells fail to increase the levels of Sec16A under the condition of cargo overload supports the notion of SNRPB (and maybe other Sm-ring components) being part of a CFFL. Thus, our data identify a role for the spliceosome in the regulation of ER-proteostasis.

Mutations in SNRPB were shown to be linked to a rare genetic disease called Cerebro-costo-mandibular syndrome (CCMS).

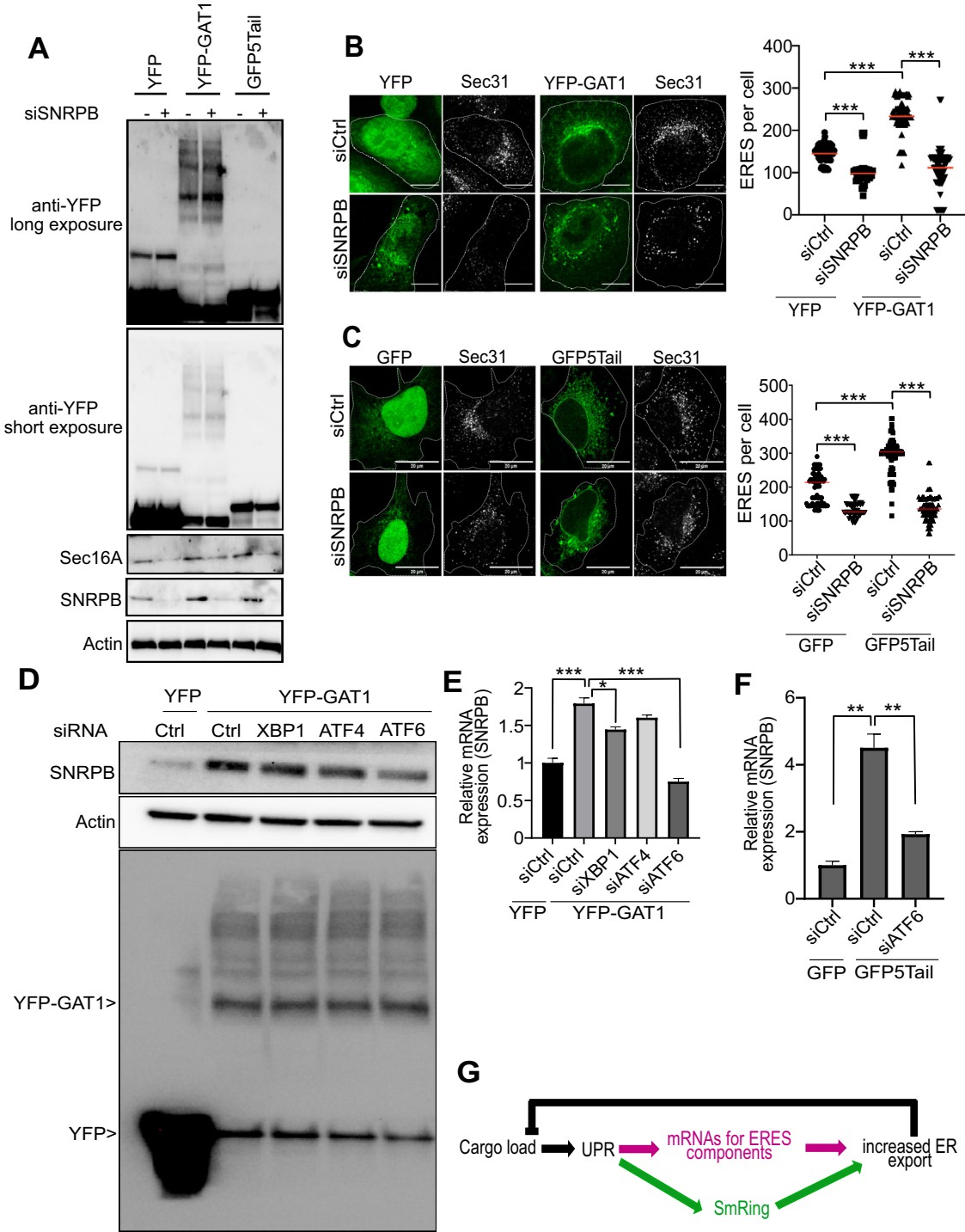

Skeletal abnormalities are a hallmark of CCMS, but there was no obvious link between mutations in a spliceosomal component and bone abnormalities. Our results provide mechanistic insights into the pathophysiology of the disease. Based on our observations, we propose that the mechanism of bone deformations is due to a perturbation of ER homeostasis and the trafficking of collagen-I. Previous work has demonstrated several examples where mutations in genes regulating ER homeostasis result in bone deformation as

has been shown for Sec23A, Sec24D, and ATF6 (Boyadjiev et al, 2006; Garbes et al, 2015; Guo et al, 2016). Further support for this notion comes from our experiment in patient-derived fibroblasts where we could increase the number of ERES by expressing wild type SNRPB. We are aware that it is difficult to deduce clinically relevant effects based on cell culture experiments, but this SNRPB-ERES link opens the possibility that a defect in ER export might be associated with bone developmental defects in CCMS patients.

**Figure 7. SNRPB is regulated by cargo load.**

(A) HeLa cells were transfected with siRNA against SNRPB (+) or with a non-targeting control siRNA (−). After 48 h, cells were transfected with a plasmid encoding YFP or YFP-tagged GAT1 or GFP-b(5)tail (GFP5Tail). After an additional 24 h, cells were lysed and immunoblotted as indicated. Scale bar = 10 μm. (B) HeLa cells were transfected with siRNA against SNRPB (siSNRPB) or with a non-targeting control siRNA (siCtrl). After 48 h, cells were transfected with a plasmid encoding YFP or YFP-tagged GAT1. After an additional 24 h, cells were fixed and immunostained against Sec31 to label ERES. The number of ERES was determined using ImageJ and the result of three independent experiments is displayed in the graph to the right. ***Indicates a *P* value < 0.001 obtained using one-way ANOVA. (C) Same experimental setup as in (B), except that cells were lysed transfected with GFP-b(5)tail. ***Indicates a *P* value smaller than 0.001 obtained using one-way ANOVA. (D) HeLa cells were transfected with siRNA against the indicated genes, or with a non-targeting control siRNA (siCtrl). After 48 h, cells were transfected with a plasmid encoding YFP or YFP-tagged GAT1. After an additional 24 h, cells were lysed and immunoblotted as indicated. (E) Same as in (D), except that we performed qPCR instead of an immunoblot. The graph is an average of three independent experiments ±SD and shows the levels of SNPRB mRNA normalized to siCtrl. ***Indicates a *P* value < 0.001, and * indicates a *P* values < 0.05, obtained using one-way ANOVA. (F), HeLa cells were transfected with siRNA against the indicated genes, or with a non-targeting control siRNA (siCtrl). After 48 h, cells were transfected with a plasmid encoding GFP or GFP-5tail. After an additional 24 h, qPCR was performed to measure the levels of SNRPB mRNA. The graph is an average of three independent experiments ±SD and shows the levels normalized to siCtrl with GFP expression. **Indicates a *P* value < 0.01 obtained using one-way ANOVA. (G) Schematic depicting the role of SNRPB as apart of a feed-forward loop in ER proteostasis. Source data are available online for this figure.

Altogether, our work establishes a regulatory role for the SNRPB and other Sm-ring components in ER-proteostasis and provides insights into the pathophysiology of Cerebro-costo-mandibular syndrome.

# Methods

## Cell culture and transfection

HeLa cells, U2OS and HeLa-Rush cells (stably expressing ManII-GFP RUSH plasmid) were cultured at 37 °C, 5% $CO_2$, and normal humidity. For confocal microscopy, cells were seeded at a density of $2 \times 10^5$ cells per well in a six-well plate. All siRNAs used in this study are listed in the key resource table. Depletions were performed with siRNA from ThermoFisher Scientific (silencer select) and independently validated with siRNA from Dharmacon (Dharmacon siGenome-SMARTpool). The non-targeting siRNA from Thermofisher Scientific (silencer select) was used as the siRNA control. Briefly, 10 nM siRNAs were mixed with 6 μl of HiPerFect transfection reagent (Qiagen, #301707) in 100 μl of serum-free DMEM and added to freshly plated cells drop by drop. Plasmids were transiently overexpressed using TransIT-LT1 (Mirus, MIR2306) according to the instructions of the manufacturer. In short, 750 ng of DNA was transfected per well, in a six-well plate for 1 day using 3 μl of Mirus per 1 μg of DNA. For collagen-1 experiments, 50 ng/ml of ascorbic acid was treated for the indicated time points. Transfection of fibroblasts (described in Bacrot et al, 2015) was performed by electroporation using a Neon NxT electroporation system (ThermoFisher Scientific).

## Immunofluorescence staining and confocal microscopy

Cells grown on glass coverslips were fixed in 4% PFA for 6 min, washed with PBS, incubated in 50 mM $NH_4Cl$ for 8 min at room temperature and then washed with PBS. Next, the cells were incubated in permeabilization buffer (0.25% Triton X-100 in PBS) for 8 min at room temperature, washed with washing buffer (0.05% tween-20) and then incubated with blocking buffer (5% BSA in washing buffer) for 1 h at room temperature. Cells were incubated with primary antibodies in 1% BSA (in washing buffer) for 1 h at room temperature. After this incubation, the cells were washed with washing buffer and incubated for 1 h with secondary antibodies in 1% BSA. The coverslips were briefly rinsed with Milli-Q water and embedded in polyvinyl alcohol mounting

medium (Sigma-Aldrich, catalog # 10981). Fluorescence images were acquired on a LSM 700 confocal microscope using a PlanAphochromat 63×/1.40 Oil Ph3 M27 objective. For ERES experiments, images were taken, focusing at nuclear ring and adjust the threshold level, when only puncta signal is detected. The number of ERES were counted per cell. The list of all antibodies including dilutions is provided under Key Resources below.

## Immunoblotting

Briefly, cells were lysed in RIPA buffer supplemented with cocktail of protease and phosphatase inhibitors (Theremofisher; A32961). The blots were incubated with primary antibodies overnight in cold room, washed with PBST and subsequently incubated with HRP secondary antibodies for 1 h at room temperature.

## Analysis of ERGIC-53-positive puncta and ERES

The analysis was performed in a manner similar to the analysis of ERES as described in our previous publications (Phuyal et al, 2022) and the analysis of peripheral ERGIC puncta was done in a similar manner. Briefly, First the image is thresholded such that ERGIC-53 positive structures are included. Then, we count only structures that are smaller than 10 pixel (to exlude the central region of ERGIC-53), but bigger than 0.5 pixels (to avoid counting noise). An example of how counting peripheral ERGIC structures was conducted is displayed in Appendix Fig. S11, which shows the counts for the cells shown in Fig. 1C.

## RNA isolation and qPCR

To measure mRNA, cells were transfected with siRNAs/plasmid as described earlier and RNA was extracted using GenElute™ Mammalian Total RNA Miniprep Kit (Sigma-Aldrich, # RTN70). RNA was reverse transcribed using High-Capacity cDNA Reverse Transcription Kit (ThermoFisher #4368814). Real-time PCR was performed with 9 ng of cDNA per reaction using Quantitect primers (Qiagen).

## Nuclear fractionation

Cells were incubated with hypotonic solution containing Hepes (pH 7.5, 10 mM), MgCl2 (2 mM), KCl (25 mM) and cocktail of protease/phosphatase inhibitors for 1 h. Cell were broken using

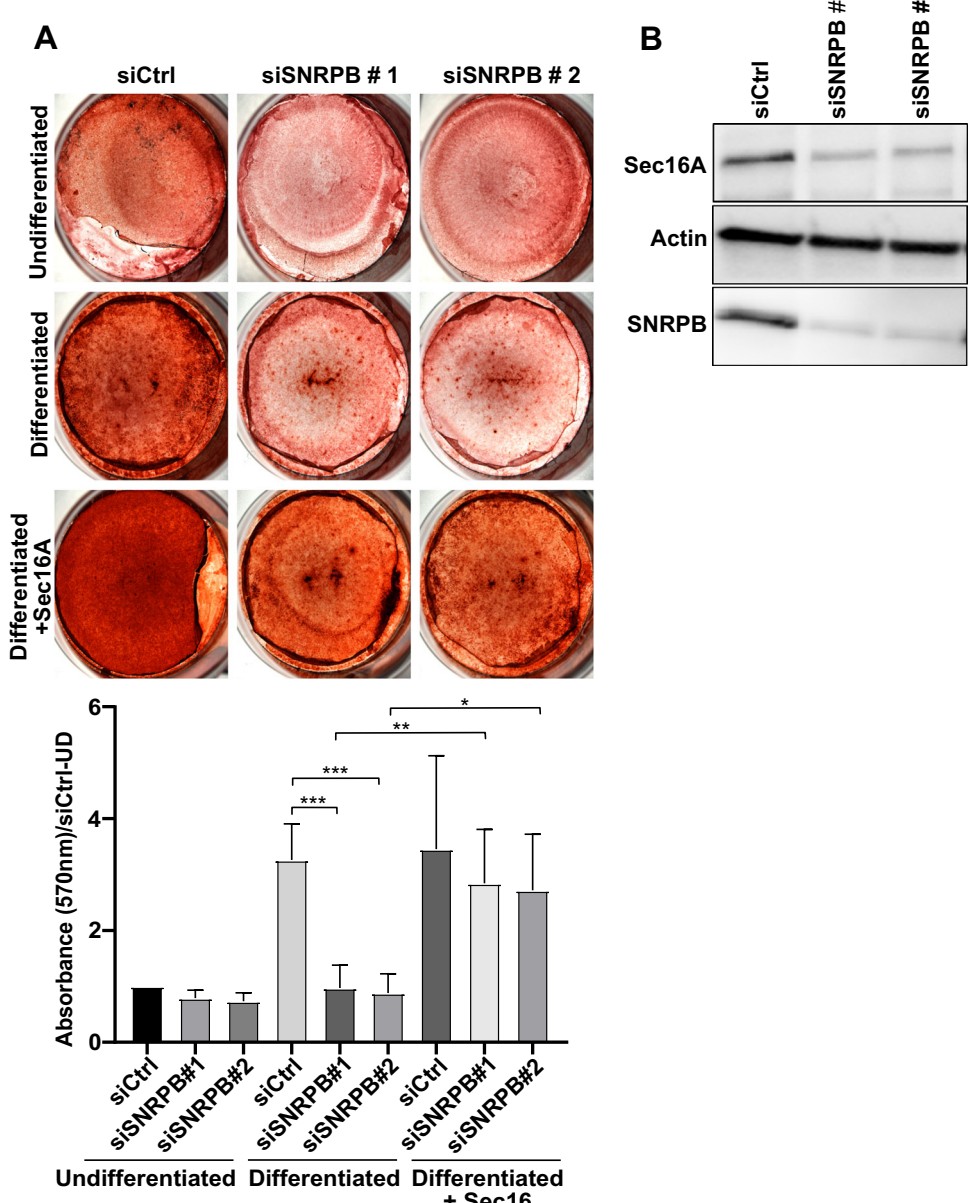

**Figure 8. Loss of SNRPB results in osteogenesis defect in vitro, which is rescued by Sec16A.**

(A) Mesenchymal cells were transfected with non-targeting siRNA (Ctrl) or with two siRNAs against SNRPB. Cells were subjected to osteogenic differentiation protocol for 9 days followed by alizarin red staining. The graph at the bottom shows quantification of three independent experiments ±SD. Absorbance at 570 nm was measured and all values are normalized as fold of siCtrl in the undifferentiated condition. Statistical testing was performed using one-way ANOVA. *$P < 0.05$, **$P < 0.01$, ***$P < 0.001$. (B) Demonstration of knockdown efficiency of the SNRPB siRNAs in mesenchymal cells using immunoblotting (B). Data are from three independent experiments ±SD. Source data are available online for this figure.

Pellet Pestle Motor Kontes (Thomas Scientific) and Disposable Polypropylene, RNase-Free Pellet Pestles (Thomas Scientific) for 15 s with a break every 5 s. The cells were checked under light microscope for nuclear integrity. After homogenization, 2 M sucrose solution was added at concentration of 125 μl/ml and then centrifuged at 600×*g* for 10 min in a swinging bucket router. Pellet contains nuclei and are washed twice with hypotonic solution containing sucrose. Pellet and supernatant are used for RNA solution same as above. The whole protocol is performed at 4 °C.

## RNA-Seq analysis

Libraries were sequenced on an Illumina HiSeq3000 flow cell, using 2-lanes. This yielded ~35–40 million paired-end 150 nt reads for biological triplicates and technical duplicates of siCTRL and siSNRPB samples. These were analyzed for gene expression and alternative splicing using independent pipelines. For differential gene expression reads were aligned to the human genome using salmon (version 1.5.1) (Patro et al, 2017). Salmon quantifications of gene expression were

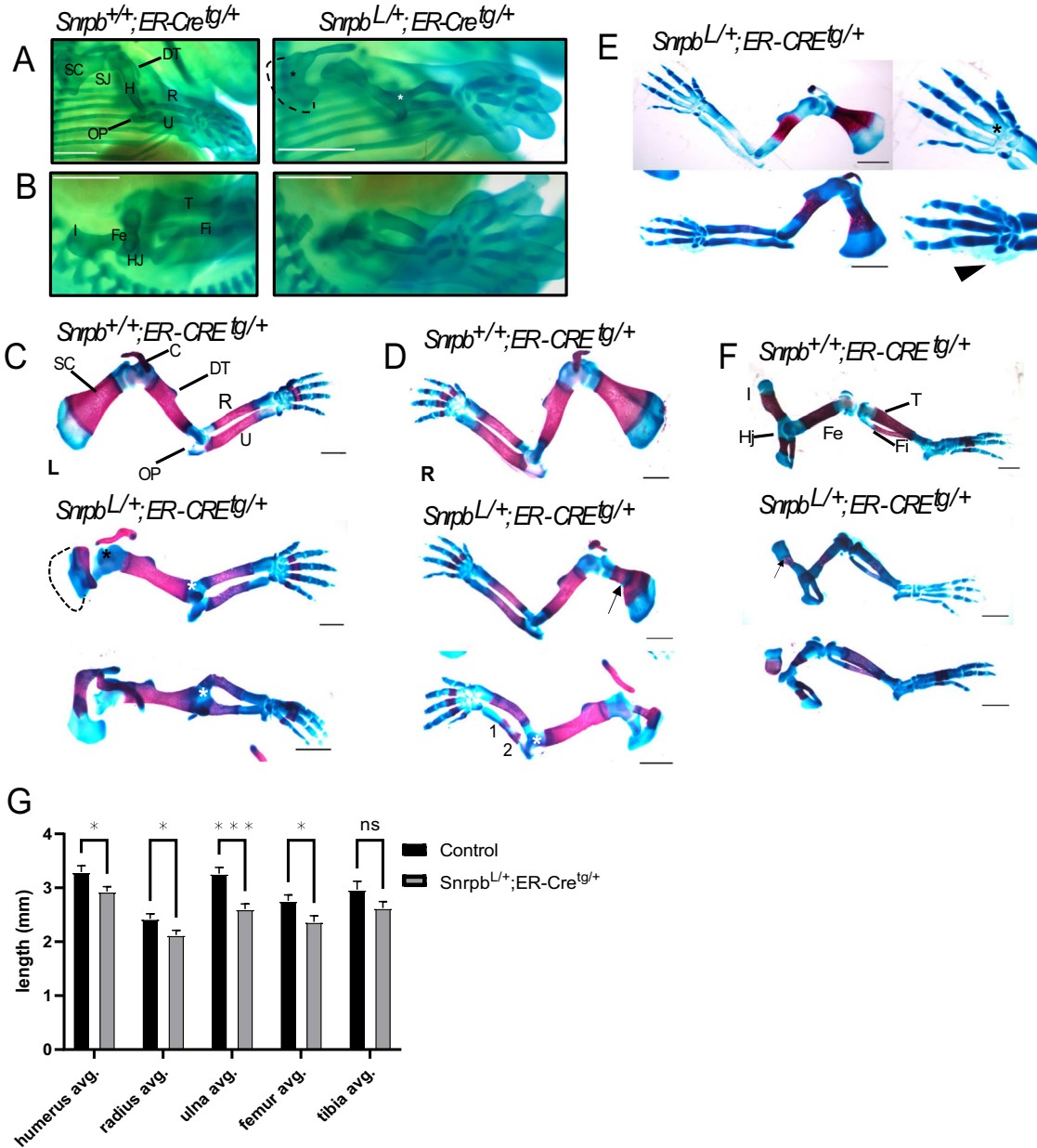

**Figure 9. Tamoxifen-induced *Snrpb* deletion before E8.5 causes chondrogenesis and osteogenesis defects in appendages.**

Representative images of Alcian Blue-strained E14.5 control (*Snrpb^+/+^; ER-Cre^tg/+^*) and mutant (*Snrpb^L/+^; ER-Cre^tg/+^*) embryos showing chondrogenesis defects in (**A**) forelimbs and (**B**) hindlimbs. (**A**) Dorsal-lateral view of right forelimbs showing missing shoulder blade (dashed line), fused shoulder joint (SJ, black asterisk), narrow humerus (H), absent deltoid tuberosity (DT), fused elbow joint (EJ, white asterisk), malformed olecranon process (OP), and bent radius (R) in the mutant. (**B**) Dorsal-lateral view of right hindlimbs showing missing iliac bone (I), narrow femur (Fe), bent tibia (T) and fibula (Fi) in the mutant. Representative images of Alcian Blue- and Alizarin Red-stained E17.5 control and mutant embryos showing chondrogenesis and osteogenesis defects in the left (**C**) and right (**D, E**) forelimbs and the right hindlimbs (**F**). (**C**) Ventral-lateral view of left forelimbs showing missing scapula blade (dashed line), curved clavicle (C), fused shoulder joint (black asterisk), misshapen deltoid tuberosity, fused elbow joint (white asterisk), malformed olecranon process, bent radius and ulna (U) in mutants. (**D**) Ventral-lateral view of right forelimbs showing smaller and narrower scapula (SC, arrow), bifurcated blade (double curved arrow), curved clavicle (C), misshapen deltoid tuberosity, fused elbow joint (white asterisk), malformed olecranon process, bent radius, and abnormal ulna ossification (two domains labeled 1 and 2) in mutants. (**E**) Ventral-lateral view of right forelimbs showing smaller and narrower scapula, missing Alizarin red staining in the radius and ulna (top), fused 2nd and 3rd metacarpals (black asterisk), clinodactyly (arrow), and missing 5th digit (arrowhead) in mutants. (**F**) Dorsal-lateral view of right hindlimbs showing thin (top, arrow) and missing iliac bone (bottom), narrow femur (bottom), and bent tibia (bottom) in mutants. Scale bar = 1 mm. (**G**) Length of the humerus, radius, ulna, and femur is reduced in *Snrpb^L/+^; ER-Cre^tg/+^* mutants (*n* = 12) compared to controls (*n* = 15) (unpaired, two-tailed *t* test). \*$P < 0.05$; \*\*\*$P < 0.0001$. Error bars indicate SEM. Source data are available online for this figure.

imported into RStudio using tximport and analyzed for differential gene expression using Deseq2 (version 1.12.3) (Love et al, 2014). Genes associated with the GO terms "COPII-coated vesicle budding" and "regulation of regulated secretory pathway" were downloaded from Panther GO and merged with the Deseq2 output using standard Python code. For alternative splicing two independent splicing pipelines were used. First, reads were aligned to the human hg38 genome, using STAR (version 2.7.9a) (Dobin et al, 2013), yielding ~85% uniquely aligned reads. Alternative splicing changes were calculated using rMATS (version v3.1.0) (Shen et al, 2014). To obtain only high confidence targets, only targets with a P value < 0.01 and a deltaPSI > 0.15 were considered alternatively spliced. In addition, to filter out splicing events in weakly expressed genes or gene regions with low expression, events with less than 100 combined junction reads in all samples were excluded. Second, alternative splicing changes were quantified—independent rom the STAR/RMATS pipeline—using Whippet (v0.11.1) (Sterne-Weiler et al, 2018). High confidence targets were filtered on Whippet-delta derived Probability > 0.95 and a deltaPSI > 0.15. To investigate exon features, exon and intron length were calculated using standard Python code. Splice site coordinates were extracted from the rMATS output table in bed format ($-20$ to $+3$ for 3'-ss; $-3$ to $+6$ for 5'-ss) and splice site sequences were extracted using bedtools. Sequences were input to MaxEntScan (Yeo and Burge, 2004), using the Maximum Entropy Model to score. RNA-seq data from the siRNA knockdowns been deposited to Gene Expression Omnibus (GEO) and are accessible under accession number GSE254122 (Reviewer Access Token: slmbckmmnbgrdyh).

## Osteogenic differentiation

MC3T3-E1 cells were cultured in MEM-α media, supplemented with 10% FBS under normal humidity and 5% $CO_2$. For the osteogenic differentiation of MC3T3, cells were cultured in osteogenic media (DMEM media supplemented with 10% FBS, 50 μg/ml of ascorbic acid and 1 mM β-glycerophosphate for 18 days. The media were changed every 3 days. For Alizrin red staining, cells were fixed with PFA (4%) and gently washed with $dH_2O$. The cells were then incubated with 40 mM (pH 4.3) alizarin red for 2 h at room temperature and then washed with dH2O with gentle shaking. For the quantification of the alizarin red staining, the wells were incubated with 35 mg/ml of Cetylpyridinium Chloride (CPC) for 2 h at 37 °C until fully dissolved. The solution was collected, cleared by brief centrifugation, diluted in CPC, and then measured at 570 nm.

For the knockdown experiments, plates were coated with siRNA using Lipofectamine 2000 as transfection reagent. After 24 h, GFP-Sec16A plasmid was transfected and cells were allowed to express the protein for 24 h. Afterward, media were changed to differentiation media and cells were allowed to differentiate for 18 days followed by Alizarin red staining as indicated above.

## Mouse lines

All procedures and experiments were performed according to the guidelines of the Canadian Council on Animal Care and approved by the Animal Care Committee of the Montreal Children's Hospital. Wnt1-Cre2 (Lewis et al, 2013) mice on the C57BL/6J genetic background were purchased from The Jackson Laboratory (strain#: 022501). The development of a conditional knock-out

Snrpb allele with CRISPR/Cas9-mediated homology-directed repair (HDR) strategy was previously described (Alam et al, 2022).

## Generation of neural crest cell-specific Snrpb$^{+/-}$ mutant embryos (Snrpb$^{ncc+/-}$)

To generate embryos with neural crest-specific Snrpb heterozygosity, Wnt1-Cre2$^{tg/+}$ animals were mated with Snrpb$^{loxp/+}$ mice. Embryos obtained from these mating were Snrpb heterozygous mutants in the neural crest cells and their derivatives, while all other cells were Snrpb wild-type.

## Collection and genotyping of embryos

The day of the plug was considered embryonic day 0.5 (E0.5). Embryos were collected at E9.5. On the day of dissection, embryos were removed from their extraembryonic membranes and assessed for the presence of a heartbeat. The somite number was counted under a light microscope (Leica MZ6 Infinity1 stereomicroscope) at the time of dissection. Embryos were fixed in 4% paraformaldehyde in PBS at 4 °C overnight, washed, and kept in PBS at 4 °C until use. Yolk sacs were used for genomic DNA extraction and genotyping (Hou et al, 2017). Genotyping to identify Snrpb wild-type and conditional alleles (with loxP sequences) was previously described (Alam et al, 2022). The primers used for the genotyping were: forward—5'-CCCGAGACAGACACAACATAAG-3', reverse—5'-GCTTTGAAGGTCCCGATGAA-3'.

## Tamoxifen treatment

In the morning of E8.5, pregnant females were injected with a single dose Tamoxifen (Sigma, T5648), 2 mg per 20 g mice, intraperitoneally. Tamoxifen was dissolved at a concentration of 10 mg/ml in sterile corn oil.

## Cartilage and skeletal preparation of embryos

To investigate cartilage formation, embryos were stained with Alcian Blue. For skeletal staining, the skin was removed from freshly dissected E17.5 embryos and neonatal pups and stained as described by (Wallin et al, 1994). Cartilage and skeletal preparations were imaged and analyzed using a Lecia stereomicroscope (Lecia MZ6). Measurements were taken using Fiji ImageJ and statistical analysis was carried out on Graphpad, Prism (https://www.graphpad.com/scientific-software/prism/).

## Preparation of embryos for embedding and histology

For cryo-embedding, fixed embryos were first cryoprotected in 30% sucrose overnight, embedded in cryomatrix, and sectioned at 10 mm thickness for immunofluorescence.

## Immunofluorescence of tissue section

After three 10-min washes with phosphate-buffered saline (PBS), slides were microwaved for 1 min at 100% power and 3 min at 20% power in 10 mM Na Citrate (pH 6) for antigen-retrieval. Once returned to room temperature, slides were blocked at 4 °C overnight with blocking buffer, 10% horse serum in PBS containing

0.25% Triton X-100. Slides were then incubated with Anti-Type I Collagen (1:100; SouthernBiotech; Cat. # 310-01) primary antibody, diluted in blocking buffer, at 4 °C overnight. After washes with PBS, slides were incubated with fluorescein-conjugated secondary antibody Rabbit anti-Goat IgG (H + L) Cross-Adsorbed Secondary Antibody, Alexa Fluor™ 488 (1:500; Invitrogen; Cat. # A-11078), diluted in blocking buffer, at room temperature for an hour. After washes with PBS, slides were mounted with FluoroshieldTM with DAPI (Sigma-Aldrich; F6057) to visualize the nuclei. Confocal images were obtained using Zeiss LSM880 laser scanning confocal microscope.

### Key resources used
### Antibodies

| | | |
|---|---|---|
| Streptavidin, Alexa Fluor™ 647 conjugate | ThermoFisher Scientific | S21374 |
| Relia Blot HRP conjugate | Bethyl Laboratories | WB113 |
| Reiia HRP-goat anti-mouse | Bethyl Laboratories | WB120 |
| Pierce™ High Sensitivity Streptavidin-HRP | ThermoFisher Scientific | 21130 |
| Peroxidase-AffiniPure Goat Anti-Mouse IgG (H + L) | Jackson ImmunoResearch | 115-035-003 |
| Peroxidase-AffiniPure Goat Anti-Mouse IgG (H + L) | Jackson ImmunoResearch | 115-035-003 |
| goat anti-human-HRP | Santa Cruz | sc-2453 |
| donkey anti-goat CFL 487 | Santa Cruz | sc-362255 |
| Anti-β-catenin | Santa Cruz | sc-7963 |
| Anti-v-SNARE Ykt6p (E-2) | Santa Cruz Biotechnology | sc-365732 |
| Anti-SNRPB (Y12) | ThermoFisher Scientific | MA513449 |
| Anti-Sec31 | BD | 612351 |
| Anti-SEC24C | Bethyl Laboratories | A304-759A |
| Anti-SEC24D | Bethyl Laboratories | A304-759A |
| Anti-SEC23B | ThermoFisher Scientific | PA531589 |
| Anti-Sec16A (KIAA0310) | Bethyl Laboratories | A300-648A |
| Anti-SEC13 | R&D Systems | MAB9055-100 |
| Anti-Sar1 | Abcam | ab125871 |
| Anti-Rtn-3 (F-6) | Santa Cruz Biotechnology | sc-374599 |
| Anti-PREB (Sec12) | Proteintech | 10146-2-AP |
| Anti-GOSR1(HFD9) | Novus Biologicals | NBP1-97437 |
| Anti-Climp63 | In house (Rabbit) | |
| Anti-GM130 | BD | 610823 |
| Anti-Giantin | Abcam | ab80864 |
| Anti-GFP (E385) | Abcam | ab32146 |

| | | |
|---|---|---|
| AntiERGIC53 | In house (Mouse Monoclonal) | G1/093 |
| Anti-Calnexin (E-10) | Santa Cruz Biotechnology | sc-46669 |
| Anti-beta-Actin (C4) | Santa Cruz Biotechnology | sc-47778 |

### Chemicals

| | | |
|---|---|---|
| AMPICILLIN ANHYDROUS CRYSTALLINE | Sigma-Aldrich | A9393-5G |
| Biotin | Sigma | G7651 |
| CYCLOHEXIMIDE READY MADE | Sigma-Aldrich | C4859-1ML |
| FORMALDEHYDE MOLECULAR BIOLOGY REAGENT | Sigma-Aldrich | F8775-25ML |
| KANAMYCIN DISULFATE SALT | Sigma-Aldrich | K1876-1G |
| PARAFORMALDEHYDE POWDER 95% | Sigma-Aldrich | 158127-500G |
| Pierce 660 nm protein assay reagent | Thermo Scientific | 22660 |
| SAPONIN FROM QUILLAJA BARK | Sigma-Aldrich | S7900-25G |
| Tris | Roche | 10708976001 |
| Bortezomib | Selleckchem | S1013 |
| Cover glasses, round 13 mm No. 1½ | VWR | 630-2191 |
| dNTP Mix (25 mM each) | ThermoFisher Scientific | R1121 |
| ENDOPLASMIC RETICULUM ISOLATION KIT | Sigma-Aldrich | ER0100-1KT |
| GeneRuler 1 kb DNA Ladder, ready-to-use | Thermo Scientific | SM0313 |
| ReliaBLOT® IP/Western Blot Reagents, WB120 | Bethyl Laboratories | WB120 |
| SensiFast SYBR Hi-ROX | Bioline | BIO-920055 |
| Tris-(hydroksymetyl)aminometan | VWR | 33621.260 |
| Trypsin-EDTA (0.25%), Phenol red | Fisher Scientific AS | 25200072 |
| PageRuler™ Plus Prestained Protein Ladder, 10 to 250 kDa | ThermoFisher Scientific | 26620 |
| Phusion High-Fidelity PCR Master Mix with HF Buffer | ThermoFisher Scientific | F531S |
| dNTP Mix (10 mM each) | ThermoFisher Scientific | R0192 |
| High-Capacity cDNA Reverse Transcription Kit (200 reactions) | Applied Biosystems/ ThermoFisher Scientific | |
| SensiFAST SYBR Hi-ROX | BioLine/Nordic Bio Site | 4368814 |

### qPCR primers

| | | |
|---|---|---|
| Hs_ACTB_2_SG QuantiTect Primer Assay | Qiagen | QT01680476 |
| | Qiagen | QT01680476 |
| | Qiagen | QT01680476 |
| Hs_CANX_1_SG QuantiTect Primer Assay | Qiagen | QT00092995 |

| | | |
|---|---|---|
| Hs_GOSR1_1_SG QuantiTect Primer Assay | Qiagen | QT00069447 |
| | Qiagen | QT00096404 |
| Hs_LOC442272_1_SG QuantiTect Primer Assay | Qiagen | QT00277718 |
| Hs_PREB_1_SG QuantiTect Primer Assay | Qiagen | QT00209958 |
| Hs_SAR1B_1_SG QuantiTect Primer Assay | Qiagen | QT00039865 |
| Hs_SEC16A_1_SG QuantiTect Primer Assay | Qiagen | QT00092778 |
| Hs_SEC23A_1_SG QuantiTect Primer Assay | Qiagen | QT00048468 |
| Hs_SEC24A_1_SG QuantiTect Primer Assay | Qiagen | QT00215859 |
| Hs_SEC24B_1_SG QuantiTect Primer Assay | Qiagen | QT00048664 |
| Hs_SEC24C_1_SG QuantiTect Primer Assay | Qiagen | QT00014336 |
| Hs_SEC24D_1_SG QuantiTect Primer Assay | Qiagen | QT00042959 |
| Hs_SNRPA1_1_SG QuantiTect Primer Assay | Qiagen | QT00094003 |
| Hs_SNRPB_1_SG QuantiTect Primer Assay | Qiagen | QT00014686 |
| Hs_SNRPB2_1_SG QuantiTect Primer Assay | Qiagen | QT00035994 |
| Hs_SNRPC_1_SG QuantiTect Primer Assay | Qiagen | QT00000707 |
| Hs_SNRPD1_1_SG QuantiTect Primer Assay | Qiagen | QT00068782 |
| Hs_SNRPE_1_SG QuantiTect Primer Assay | Qiagen | QT00217980 |
| Hs_SNRPF_1_SG QuantiTect Primer Assay | Qiagen | QT01008630 |
| Hs_SNRPG_1_SG QuantiTect Primer Assay | Qiagen | QT00200291 |
| Hs_SEC31A_1_SG QuantiTect Primer Assay | Qiagen | QT00058653 |

**Primers**

| PCR | spanning exons | cDNA [bp] | DNA[bp] | Forward primer (5´-3´) | Reverse primer (5´-3´) |
|---|---|---|---|---|---|
| Sec16A_5 | 6–8 | 331 | 952 | GGTATTGGT GTGATGCAGAGT | TGAAAGGA GCCTGGAG GAAG |
| Sec16A_11 | 17–19 | 220 | 3046 | TATGAGTAC GCCCAGTCCCT | CGAAGAG TCGTAACT GGGAAG |
| ACTB | 3–5 | 839 | 1375 | AGAAGGATT CCTATGTGGGCG | CTTGATCT TCATTGTG CTGGGTG |

**siRNAs**

| | | |
|---|---|---|
| siGenome Non-Targeting Control siRNA Pool #1 | Horizon Discovery (Dharmacon) | D-001206-13-05 |
| siGenome Smartpool_ATF4 | Dharmacon/VWR | M-004883-03-0005 |
| siGenome Smartpool_ATF6 | Dharmacon/VWR | M-009917-01-0005 |
| siGenome Smartpool_XBP1 | Dharmacon/VWR | M-009552-02-0005 |
| Silencer® Select siRNA_HS_siSNRPB #1 | ThermoFisher Scientific | 4392420 (s224659) |
| Silencer® Select siRNA_HS_siSNRPB #2 | ThermoFisher Scientific | 4427037 (s13223) |
| Silencer® Select siRNA_MM_siSNRPB (mouse) | ThermoFisher Scientific | 4390771 (s74099) |
| Silencer® Select siRNA_HS_SNRPB2 | ThermoFisher Scientific | 4427037 (s13223) |
| Silencer™ Select Negative Control No. 1 siRNA | ThermoFisher Scientific | 4390843 |
| siGenome Non-Targeting siRNA Pool #1 | Dharmacon/VWR | D-001206-13-20 |
| Silencer® Select siRNA_HS_SNRPA | ThermoFisher Scientific | 4427037 (s13213) |
| Silencer® Select siRNA_HS_SNRPC | ThermoFisher Scientific | 4427037 (s13225) |
| Silencer® Select siRNA_HS_SNRPD | ThermoFisher Scientific | 4427037 (s224661) |
| Silencer® Select siRNA_HS_SNRPE | ThermoFisher Scientific | 4427037 (s13238) |
| Silencer® Select siRNA_HS_SNRPF | ThermoFisher Scientific | 4427037 (s13241) |
| Silencer® Select siRNA_HS_SNRPG | ThermoFisher Scientific | 4427037 (s13243) |

## Data availability

The RNAseq data shown in Fig. 6 are deposited under: https://www.ncbi.nlm.nih.gov/geo/query/acc.cgi?&acc=GSE254122.

The source data of this paper are collected in the following database record: biostudies:S-SCDT-10_1038-S44318-024-00208-z.

## Peer review information

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

## Acknowledgements

The authors would like to thank Dr. Indra Gupta for her generous gifts of Anti-Type I Collagen primary antibody and Secondary Antibody. HF is supported by grants from the Norwegian Cancer Society (Project 208015), by the Research Council of Norway (Project 302452), by H2020 MSCA-innovative training networks SECRET and SAND (H2020-MSCA-ITN-860035; H2020-MSCA-ITN-859962), by grants from the Austrian Science Foundation (FWF) (Projects: P35832, P36600, FG2000). MZ is supported by a Centre of Excellence from the Norwegian Research Council (Project: 262652). UH is supported by a grant from the Austrian Science Foundation (FWF) (Project ESP634). LJM is supported by the Canadian Institutes of Health Research (CIHR), bridge funding from the Research Institute of McGill University Health Centre (RI-MUHC) and the Azrieli Foundation; Fonds de Recherche du Québec (FQRS).

## Author contributions

**Muhammad Zahoor**: Conceptualization; Formal analysis; Investigation; Methodology; Writing—original draft; Writing—review and editing. **Yanchen Dong**: Data curation; Formal analysis; Validation; Investigation; Visualization; Methodology; Writing—original draft; Writing—review and editing. **Marco Preussner**: Data curation; Formal analysis; Investigation; Visualization; Methodology. **Veronika Reiterer**: Data curation; Formal analysis; Investigation; Methodology. **Sabrina Shameen Alam**: Formal analysis; Methodology. **Margot Haun**: Investigation; Visualization; Methodology; Writing—review and editing. **Utku Horzum**: Investigation; Visualization; Methodology. **Yannick Frey**: Investigation; Visualization; Methodology. **Renata Hajdu**: Formal analysis; Methodology. **Stephan Geley**: Supervision; Methodology; Writing—original draft; Writing—review and editing. **Valérie Cormier-Daire**: Generation of patient cell line. **Florian Heyd**: Formal analysis; Supervision; Funding acquisition. **Loydie A Jerome-Majewska**: Conceptualization; Funding acquisition; Visualization; Writing—original draft; Project administration; Writing—review and editing. **Hesso Farhan**: Conceptualization; Formal analysis; Supervision; Funding acquisition; Writing—original draft; Project administration; Writing—review and editing.

Source data underlying figure panels in this paper may have individual authorship assigned. Where available, figure panel/source data authorship is listed in the following database record: biostudies:S-SCDT-10_1038-S44318-024-00208-z.

## Disclosure and competing interests statement

The authors declare no competing interests.

