## [Peer Review File · The EMBO Journal]

The unfolded protein response regulates ER exit sites via SNRPB-dependent RNA splicing and contributes to bone development.

Muhammad Zahoor, Yanchen DONG, Marco Preußner, Veronika Reiterer, Sabrina Shameen Alam, Margot Haun, Utku Horzum, Yannick Frey, Renata Hajdu, Stephan Geley, Valérie Cormier-Daire, Florian Heyd, Loydie Jerome-Majewska, and Hesso Farhan

Corresponding author(s): Hesso Farhan (hesso.farhan@i-med.ac.at) , Loydie Jerome-Majewska (loydie.majewska@mcgill.ca), Muhammad Zahoor (Mzkhaans@yahoo.com)

Review Timeline:

Submission Date:	28th Sep 23
Editorial Decision:	24th Nov 23
Revision Received:	8th Apr 24
Editorial Decision:	30th Apr 24
Revision Received:	19th Jun 24
Editorial Decision:	25th Jun 24
Revision Received:	8th Jul 24
Accepted:	24th Jul 24

Editor: Cornelius Schneider

Transaction Report:

Dear Dr. Farhan,

Thank you for submitting your manuscript for consideration by the EMBO Journal. We have now received comments from three reviewers, which are included below for your information.

As can be seen from the reports, all three referees found the results of importance and interest, agreed that the experiments were performed competently but both referees #1 and especially referee #2 have several concerns that require major revisions to substantiate the claims raised in the manuscript. I think it would be helpful to discuss the revision in more detail via email or phone/videoconferencing - please let me know which option you prefer. I should also add that it is The EMBO Journal policy to allow only a single major round of revision and that it is therefore important to resolve the main concerns at this stage.

We generally allow three months as standard revision time, which can be extended to 6 months in case of major revisions, such as the experiments required here. As a matter of policy, competing manuscripts published during this period will not negatively impact on our assessment of the conceptual advance presented by your study. However, we request that you contact the editor as soon as possible upon publication of any related work, to discuss how to proceed. Should you foresee a problem in meeting the deadline, please let us know in advance and we may be able to grant an extension.

Thank you for the opportunity to consider your work for publication. I look forward to your revision.

Yours sincerely,

Cornelius Schneider

Cornelius Schneider, PhD
Editor
The EMBO Journal
c.schneider@embojournal.org

Please remember: Digital image enhancement is acceptable practice, as long as it accurately represents the original data and

conforms to community standards. If a figure has been subjected to significant electronic manipulation, this must be noted in the figure legend or in the 'Materials and Methods' section. The editors reserve the right to request original versions of figures and the original images that were used to assemble the figure.

We realize that it is difficult to revise to a specific deadline. In the interest of protecting the conceptual advance provided by the work, we recommend a revision within 3 months (22nd Feb 2024). Please discuss the revision progress ahead of this time with the editor if you require more time to complete the revisions.

Referee #1:

In this work, Zahoor, Dong and colleagues propose a regulatory link between the UPR and the splicing machinery. They study the roles of SNRPB in the ER export machinery and show that SNRPB is accumulated during ER-overload, inducing increase in ER exit site number and Sec16A levels. Using an in vitro model of cerebro-costo-mandibular syndrome (CCMS), which is linked to SNRPB loss of function, they have shown that Sec16A overexpression in SNRPB silencing conditions promoted recovery of ER exit sites, indicating that bone defects in CCMS could be linked to SNRPB defects. In addition using an in vivo model of Snrpb deletion they showed that SNRBP is required to proper bone growth. The results shown in the manuscript are interesting, novel and can be helpful to further understand the role of SNRPB in CCMS and its link to the UPR machinery. Nevertheless, a deeper functional study to properly assess the link between the proteostasis machinery with ATF6 and the spliceosome with SNRPB are required. For that, some experiments are proposed below and upon revision of these points the manuscript could be considered for publication in The EMBO Journal.

General questions:

1. What other mechanisms are affected by SNRPB silencing?

a. What other known targets of SNRPB are being affected? Is it exclusive to ER Exit sites proteins?

b. There are some papers tying SNRPB with cancer-related proteins such as BRCA2 and CDK1, did the authors performed any experiments to check that the level of these proteins is not being altered? If so, how can they isolate the effects of SNRPB with ERES-related proteins and not being a spillover effect from other signalling alterations?

2. How is ATF6 inducing SNRPB? Do the authors believe that this effect is observed due to technical specifications of the UPR inducer used? (overexpression of YFP-GAT1 and GFP5tail). And why specifically ATF6 and no other members of the UPR?

Major points:

1. In figure 1A and subsequent figures, the authors should comment on how SNRPB2 siRNA was not inducing the same effect as SNRPB.

2. In figure 3B, pictures of the cells should be added to better show the effect.

3. In figure 4A, densitometry of the bands should be performed using actin as a normaliser, as the levels of actin also fluctuate among the 3 conditions. This would provide a better conclusion regarding the effects of the knockdown. Similarly, for 4B, the normalisation should be performed using actin and a graph showing fold change should be depicted. On that note, have the authors checked for off-targets of the siRNAs? Or are they verified?

4. In figure 5B, its advisable to add a nuclear exclusive and a cytosol exclusive markers to ensure the fractionation was clean.

5. The reconstitution experiment in figure 5C is crucial for the overall message of the paper, therefore it would be highly valuable to show a western blot with the protein levels for Sec16A and SNRPB.

6. Overall the panels of figure 6 need a clearer denotation and description. There are lack of legends and information in this figure.

7. In figure 7 the authors should perform experiments to show that the UPR is being induced by YFP-GAT1 and GFP5tail, such

as qPCR to measure XBP1s, BiP, CHOP, ATF6 levels and XBP1 splicing by conventional PCR.

8. Also in figure 7D the authors should perform densitometry analysis to properly measure magnitude of reduction of ATF6 levels, using actin as a normaliser.

9. It would be interesting to reconstitute ATF6 in the experiments of figure 7 and measure SNRNP levels to strengthen the conclusion of this figure.

10. What do the authors think regarding the role of S1P and S2P in SNRNP function. Could it be that ATF6 translocation to the Golgi is blocked by SNRNP silencing? In this case, could ATF6 regulate SNRNP function?

11. To better understand the role of UPR machinery and SNRNP, additional experiments checking the downstream and upstream events to ATF6 function, such as ERp18 association and ATF6f induction targets are strongly recommended.

Minor points:

1. In figure 1D it would be interesting to also show a membrane marker to better identify the borders of the cells. This would provide a stronger point regarding the localization of collagen inside or outside the cells.

2. There are typos in the methods section --Immunofluorence-- staining and confocal microscopy: For ERES experiments, images were taken, focusing at nuclear ring and adjust the threshold level, when only puncta -signal-- is detected.

Referee #2:

Zahoor et al provide interesting evidence of a role for splicing in ER proteostasis. A combination of cell and mouse-based analysis point to the levels of the Sm protein SNRNP, and other Sm proteins, being important for ER-to Golgi trafficking and levels of SNRNP being important for bone development in mice making the link to the genetic disorder CCMS. On the whole the data is convincing, well presented, and provides an interesting mechanistic explanation for the developmental defects associated with CCMS. However, there are issues with the quality of some of the data and the manuscript contains many mistakes and missing information that requires correcting. In addition, the narrative implies that it is SNRNP on its own that regulates splicing where in fact the depletion of SNRNP almost certainly affects snRNP/spliceosome function or abundance to influence splicing which is not discussed by the authors.

The manuscript starts with analysis of ER-to-Golgi trafficking using the RUSH assay with Mann and Col-X reporters following knockdown of SNRNP with siRNAs. These data convincingly show that SNRNP levels are important for ER-to-Golgi trafficking. To determine if the ER-to-Golgi trafficking of a native cargo is influenced, the localisation of ERGIC-53 was analysed following SNRNP knockdown. Unfortunately, the images and quantitation are not convincing/clear. An explanation/details of how peripheral ERGIC-53 was counted is needed somewhere. Could the images in Figure 1C be annotated to show the region of the cell considered the periphery?

Neural crest tissue from embryos with specific heterozygous neural crest cell knockout of SNRNP were next investigated for localisation of collagen-I. Unfortunately, the change in collagen-I localization implied in Figure 1D is not convincing. It is not clear how one can tell between intracellular and extracellular localisation from the presented images. Can another intracellular marker be used to show increased intracellular co-localisation of collagen-I? In contrast, the RUSH assay for collagen-I in Figure 1E reveals that SNRNP knockdown does influence collagen-I ER-to-Golgi trafficking.

To further analyze the influence of SNRNP knockdown on the number of ER exit sites (ERES), an important node in the proteostasis network, Sec31 was used as a marker. Figure 2A shows nicely that knockdown of SNRNP in HeLa cells decreases the number of ERES. The reader is next referred to Figure S1D for U2OS data but unfortunately this Figure was not included in the Supplementary materials.

In a potentially important experiment, fibroblasts from a CCMS patient were analysed for ERES numbers with and without rescue using a transfected GFP-SNRNP construct. While the quantitation presented comparing ERES numbers looks good, in the images presented in Figure 2B the Sec31 appears to be colocalising with SNRNP in the nucleus? All other non-electroporated cells have a very clearly defined nucleus without much Sec31 localization. Therefore, these images are not convincing. In addition, the information provided on the variant that the CCMS patient cells carry, 165G>C, is not informative. What type of variant is this? c.165G>C [p.Arg55Ser] would provide more information to the reader. Also, no information is given in the Methods on the source of patient fibroblasts or methods for transfection.

Now that it was established that SNRNP was required for ER-to-Golgi trafficking the authors investigated whether other Sm ring proteins were required for ER-to-Golgi trafficking. Knockdown of SNRNP, SNRPC, SNRPD1 and SNRPG was shown not to

affect the expression of each other significantly. Using the RUSH assay again to monitor ER-to-Golgi trafficking revealed that only the Sm ring proteins SNRPB, SNRPD1 and SNRPG decreased ER-to-Golgi trafficking. This result is very interesting and indicates that general disruption of snRNPs/spliceosomes, and not just SNRPB alone, disrupts ER-to-Golgi trafficking (see comments below).

The next important step in this work was to determine how reducing SNRPB levels specifically disrupted ER-to-Golgi trafficking. Levels of proteins involved in ER-to-Golgi trafficking such as Sec16A, Sar1A, Sec12, Sec24C, and Sec31 were investigated by western blotting. While there are potentially some noticeable reductions of some of these ERES proteins, these western blots (Figure 4A and EV4) would really need to be repeated, quantitated and statistical analysis carried out to bolster the significance of these results. It would also be important to see here whether knockdown of other Sm ring proteins influenced ERES protein levels. There appeared to be no influence of SNRPB knockdown on the mRNA levels of the ERES proteins.

Now, as predicted, the knockdown of SNRPB will most certainly impact the splicing of certain pre-mRNAs and most likely pre-mRNAs of ERES components that have been shown to be reduced and involved in ER-to-Golgi trafficking. Concentrating on Sec16A, a clear defect in Sec16A pre-mRNA splicing and nuclear accumulation of unspliced pre-mRNA following SNRPB knockdown was observed, whereas two other pre-mRNAs were not affected. This result is interesting and nicely supports the idea that there is a splicing defect responsible for the defects observed following SNRPB knockdown. Again, it would be important to see here whether knockdown of other Sm ring proteins influenced Sec16A splicing.

To determine how important the mis-splicing of Sec16A is to ER-to-Golgi trafficking a RUSH assay was used to determine the influence of providing exogenous Sec16A to rescue Sec16A function on ER-to-Golgi trafficking. Interesting a significant amount of ER-to-Golgi trafficking was restored by adding back Sec16A indicating that Sec16A plays a major role in ER export and is most likely one of the more important pre-mRNAs that is mis-spliced.

Global analysis of expression and splicing by RNA-seq of cells depleted of SNRPB was the logical next step. A volcano plot is presented to display differential expression and it is stated that there were not any expression changes in Sec16A or other components of the secretory pathway. It would have been appropriate here to also provide these differential expression data in a Supplementary Excel file to allow readers to see and interrogate the data themselves. Splicing analysis was carried out using two different bioinformatics tools. A typical pattern of splicing defects was observed as seen previously for depletion of other core splicing factor genes which is a change in skipped exons. Sec16A splicing was picked out to show its mis-splicing but the sashimi plot for the siSNRPB appeared not to be formatted correctly. It would have been nice here to have picked out one or two other significant/relevant mis-splices to confirm by RT-PCR and present as sashimi plots. Finally, as expected for this type of analysis, common features of pre-mRNA that displayed splicing changes were determined. Disappointingly, there is no indication that these RNA-Seq data will be made publicly available through any of the normal routes like GEO.

Next the hypothesis that SNRPB was either a mediator or effector of UPR was determined. ER overloading was used to determine the influence on SNRPB at both the mRNA and protein levels. An interesting upregulation of SNRPB is observed at the mRNA and protein levels which is determined to be ATF6 regulated. The western blots here for SNRPB levels should be repeated, quantitated and statistical analysis carried out to bolster the significance of these results. It is also important here to determine if the levels of other Sm ring protein mRNA and protein are increased following ER overload.

An important part of this work is to link the functional work here on SNRPB and ER-to-Golgi trafficking to CCMS and bone development defects seen with CCMS. The authors tested whether loss of SNRPB affects osteogenesis using an in vitro assay where mesenchymal cells were differentiated to osteoblasts. Data is clearly presented indicating that knockdown of SNRPB does in fact inhibit in vitro osteogenesis and can be rescued by Sec16A overexpression. As before it would be important to at least include knockdown of another Sm ring protein to determine if effects are specific for SNRPB.

The authors conclude by presenting very nice results in mice with induced deletion of Snrpb in E8.5 mice and looking 6 days later at bone development in the appendages. The data convincingly indicate Snrpb is required for growth and patterning of the bones of the mouse appendages further indicating the link between SNRPB and the defects associated with CCMS.

The way the manuscript is written the authors are implying that SNRPB alone is acting as a "novel node" and is regulating ER proteostasis. As SNRPB is an Sm ring component, and the authors have also provided evidence that knockdown of other Sm ring components inhibits ER-to-Golgi trafficking, the most likely hypothesis is that reduced levels of SNRPB and other Sm ring components causes either defective snRNPs/spliceosomes or reduced levels of functional snRNPs/spliceosomes to influence the splicing of a subset of pre-mRNAs required for regulating ER proteostasis. So the functional defect here is not caused by SNRPB alone but the effects of having less SNRPB have on the spliceosome. So the text should be revised throughout the manuscript to indicate that it is not SNRPB alone that is influencing ER proteostasis, but the effects of reduced SNRPB on the snRNPs/spliceosome. This point should also be emphasized in the Discussion.

Minor comments

Add to Abstract that knockdown of other Sm proteins inhibits ER-to-Golgi trafficking.

At the beginning of the third paragraph of the Introduction the first two sentences repeat the same information: "To address knowledge gap, we focused on the spliceosomal subunit SNRPB for two main reasons. To investigate this further, we focused on the spliceosomal subunit SNRPB for two reasons."

Please define ERGIC in the first paragraph of the Results.

In the third paragraph of the Results, in the first sentence, it would be useful to state here that you are using Sec31 localization as an ERES marker.

Figure 1D - please define in Figure 1 legend what the annotations in the figure "NECT" and "HM" represent.

Figure 2B legend - quantification of what? Define what Pos. and Neg. means in legend.

Figure S1 legend, first line - "siTNA" should be "siRNA"

Figure 6D legend - please indicate whether this volcano plot was produced from data derived from RMATS, Whippet or both.

Figure 8A - please provide some information in the legend as to how the quantification at the bottom of the panel was carried out.

The following sentences are not worded clearly. "We plotted the changed exons in a volcano plot in Figure 6D, again highlighting exons in ER-Golgi trafficking genes in green. Overall, we identified 40 genes that exhibited affected transcripts." Was this 40 genes overall or 40 ER-Golgi trafficking genes?

It is not clear what the siRNAs for SNRPB are. The methods say that siRNAs are listed in Supplementary Table S2 but I could not find that table. The only table I did find with siRNA information was in the "Key resources used" section which listed "siGenome Smartpool_SNRPB" and Silencer Select siRNA_HS_SNRPB". It should be added to this table which of these siRNAs are the siSNRPB#1 and siSNRPB#2 that are described in the presented data.

Overall writing style issues. Use of unqualified "This" at the start and within many sentences. Please specifically define what "This" is.

Referee #3:

In this interesting and elegant manuscript, the authors demonstrate that proteins involved in the splicing machinery are also involved in the regulation of ER proteostasis and provide a new model and mechanism in support of this. The data appear quite solid while the manuscript reads very well and the conclusions are balanced and supported by the experimental evidence. I only have a few minor comments:

1. In the second paragraph of the Introduction, there is a repeated sentence: "To address knowledge gap, we focused on the spliceosomal subunit SNRPB for two main reasons." should be deleted.
2. In the first section of the results, there is a reference to Figure S1D but this does not exist. I think it should say Figure 1C. Moreover, it is unclear why supplementary figures are labeled as EV1, EV2, EV3, etc....
3. It is a bit unclear why in Figure 5A there appears to be multiple bands in the Sec16A without the siSNRPB. The authors describe this as potential splicing intermediates, but why they don't show in the other PCR reactions (for CANX and ACTB)? I guess it has to do with the primer annealing locations for each of those targets.... This is just a minor comment and the author may address it if they wish.
4. In Figure 9AB the labeling is barely readable and I would suggest to increase the fonts of those abbreviations.
5. In the last section of the results, the word 'pelvic' should read pelvis ('In the pelvic, the iliac bone was thinner....').

Referee #1:

In this work, Zahoor, Dong and colleagues propose a regulatory link between the UPR and the splicing machinery. They study the roles of SNRPB in the ER export machinery and show that SNRPB is accumulated during ER-overload, inducing increase in ER exit site number and Sec16A levels. Using an in vitro model of cerebro-costo-mandibular syndrome (CCMS), which is linked to SNRPB loss of function, they have shown that Sec16A overexpression in SNRPB silencing conditions promoted recovery of ER exit sites, indicating that bone defects in CCMS could be linked to SNRPB defects. In addition using an in vivo model of Snrpb deletion they showed that SNRBP is required to proper bone growth. The results shown in the manuscript are interesting, novel and can be helpful to further understand the role of SNRPB in CCMS and its link to the UPR machinery. Nevertheless, a deeper functional study to properly assess the link between the proteostasis machinery with ATF6 and the spliceosome with SNRPB are required. For that, some experiments are proposed below and upon revision of these points the manuscript could be considered for publication in The EMBO Journal.

Response: we thank the reviewer for the constructive and supportive comments of our manuscript.

General questions:

1. What other mechanisms are affected by SNRPB silencing?

a. What other known targets of SNRPB are being affected? Is it exclusive to ER Exit sites proteins?

b. There are some papers tying SNRPB with cancer-related proteins such as BRCA2 and CDK1, did the authors performed any experiments to check that the level of these proteins is not being altered? If so, how can they isolate the effects of SNRPB with ERES-related proteins and not being a spillover effect from other signalling alterations?

Response: SNRPB does not affect only ERES proteins. The focus of our work was on ERES, which is why we concentrated our efforts on Sec16A and COPII components. It is very likely that SNRPB affects a wide range of proteins, but the point of our work is that its effect on ERES is what is relevant for collagen trafficking. This is also shown in our in vitro osteogenesis assay where we show that overexpression of Sec16A rescues the osteogenesis defect of SNRPB knockdown cells. SNRPB was also previously shown to affect other proteins of the secretory pathway, such as Rab26 (<https://www.nature.com/articles/s41419-019-1929-y>). Although this paper did not investigate membrane trafficking, Rab26 is known to regulate the Golgi export of a subset of proteins. We are aware of the fact that SNRPB affects proteins like CDK1. In fact, CDK1 is known to phosphorylate Rab1A, a regulator of ER-to-Golgi trafficking. This phosphorylation (to the best of our knowledge) is part of the mitotic diassembly of ERES (PMID: 1902553; PMID: 27041585). A reduction of CDK1 would result in a G2-phase arrest which is not known to have to any effect on ERES. Moreover, the lack of a negative effect on Rab1A, would be expected to accelerate trafficking, while we actually see a reduction in trafficking. In addition, we would like to point to the fact that the defect in ER-Golgi trafficking could be rescued by Sec16A overexpression, which shows this effect is not related to Rab1A.

2. How is ATF6 inducing SNRPB? Do the authors believe that this effect is observed due to technical specifications of the UPR inducer used? (overexpression of YFP-GAT1 and GFP5tail). And why specifically ATF6 and no other members of the UPR?

Response: We don't think that these conditions are specific to ATF6. Overexpression of GAT1 was shown in our previous work (Farhan et al, EMBOJ, 2008) to regulate the number of ERES in a manner dependent on IRE1. There is no obvious reason why overexpression of secretory cargo would activate only one branch of the UPR. We think that more than one branches of the UPR are activated, but that the induction of SNRPB appears to be mostly dependent on ATF6. This is supported by the observation that silencing ATF6 prevents SNRPB induction in response to cargo overload (Figure 7D). In addition, our new data showing that ERp18 knockdown also prevents SNRPB induction by cargo overload (supplementary Figure EV9).

Major points:

1. In figure 1A and subsequent figures, the authors should comment on how SNRPB2 siRNA was not inducing the same effect as SNRPB.

Response: SNRPB2 is not part of the Sm-ring. We appreciate that the nomenclature might be misleading, because SNRPB2 could be mistaken for SNRPB#2, which is a second siRNA against SNRPB. However, we do not know how to address this properly.

2. In figure 3B, pictures of the cells should be added to better show the effect.

Response: the images have been added and are now displayed as the new Figure 3A. The former Figure 3A is now supplementary Figure EV4.

3. In figure 4A, densitometry of the bands should be performed using actin as a normaliser, as the levels of actin also fluctuate among the 3 conditions. This would provide a better conclusion regarding the effects of the knockdown. Similarly, for 4B, the normalisation should be performed using actin and a graph showing fold change should be depicted. On that note, have the authors checked for off-targets of the siRNAs? Or are they verified?

Response: densitometries were added to Figure 4. The siRNAs are not validated by the company, but we checked for knockdown efficiency and we used two different non-overlapping siRNAs.

As for the qPCR in Figure 4B, this is normalized internally against GAPDH. We think it is more appropriate to show the fold change versus the control condition.

4. In figure 5B, its advisable to add a nuclear exclusive and a cytosol exclusive markers to ensure the fractionation was clean.

Response: We performed the fractionation and added a figure that shows the quality of the fractionation protocol. This is now shown as Figure 3C.

5. The reconstitution experiment in figure 5C is crucial for the overall message of the paper, therefore it would be highly valuable to show a western blot with the protein levels for Sec16A and SNRPB.

Response: We repeated this experiment to produce the control that the reviewer asked for. We show now in Figure 5D (previously 5C) that GFP-Sec16A as well as GFP are expressed and also that these overexpressions do not affect the knockdown efficiency. We think the latter is another very useful control, because knockdowns tend to become leaky when cells are transfected with plasmids.

6. Overall the panels of figure 6 need a clearer denotation and description. There are lack of legends and information in this figure.

Response: The quality of the Sashimi plot in Figure 6 was improved. We also added more information to the legend.

7. In figure 7 the authors should perform experiments to show that the UPR is being induced by YFP-GAT1 and GFP5tail, such as qPCR to measure XBP1s, BiP, CHOP, ATF6 levels and XBP1 splicing by conventional PCR.

Response: The reason why we chose to use secretory protein overexpression rather than chemical drugs like thapsigargin, is that we think that they represent more physiological inducers of the UPR. In a previous paper, we showed that GAT1 Overexpression induces XBP1s (Farhan et al, EMBOJ, 2008), but we never tested ATF6. We used a luciferase reporter for ATF6 activation and found that while thapsigargin led to a more than 10fold induction of the ATF6 reporter activity, GAT1 overexpression only led to a 1.5 fold induction. This is as expected much lower than the chemical UPR induction. We include these data (performed in triplicates) in this letter to the reviewer. We can also add them to the manuscript, but would like to not include this piece of data. The reason is that they distract from the main point of the manuscript. They open the discussion about “how much UPR is a physiological UPR?” or “what is the amplitude of ATF6 activation necessary to activate ERES?”. These are very important questions, but we feel that they belong to a separate story, and distract from our major findings, which is a connection between that spliceosome and ERES that involved the ATF6 branch of the UPR. Our evidence that ATF6 is involved is based on: (i) silencing ATF6 blocks the induction of SNRPB by cargo overload and (ii) silencing ERp18 also blocks this induction, again supporting a role for ATF6. We leave the decision to the reviewer to decide whether the data below should be included into the manuscript.

We also tested a compound that has been proposed to and is sold as an ATF6 specific activator (AA147). However, we did not observe any activation of ATF6 by this drug at all, as seen in the figure below where we show that thapsigargin induces ATF6 translocation to the Golgi, while AA147 looks like control cells. This was

done with endogenous ATF6. We do not know why we are not able to reproduce these data from the literature, but we therefore cannot use this drug for our current work.

Thapsigargin final concentration is 2uM for 6h
AA147 final concentration 10uM for 24h

8. Also in figure 7D the authors should perform densitometry analysis to properly measure magnitude of reduction of ATF6 levels, using actin as a normaliser.

Response: densitometry was made and is now shown as the new supplementary Figure EV9.

9. It would be interesting to reconstitute ATF6 in the experiments of figure 7 and measure SNRPB levels to strengthen the conclusion of this figure.

Response: We guess that the reviewer meant that we should overexpress ATF6 and determine its effect on SNRPB. However, the problem with this experiment is that overexpression of ATF6 will not necessarily mean that ATF6 is active (unless induced by an ER stressor). We tried to follow the reviewer's suggestion and purchased a compound that is supposed to act as an ATF6 activator (AA147). Surprisingly, in our hands, this compound had no effect an ATF6 activation at any concentration tested.

10. What do the authors think regarding the role of S1P and S2P in SNRPB function. Could it be that ATF6 translocation to the Golgi is blocked by SNRPB silencing? In this case, could ATF6 regulate SNRPB function?

Response: We have not analyzed the activities or localization of SP1 and SP2. We think that localization and activity of SP1/2 should not be affected by SNRPB depletion.

We assume the question of the reviewer is whether SNRPB regulates ATF6, but controlling its trafficking to the Golgi. We assume that this might indeed be the case. Silencing SNRPB would, by affecting ER-export in general, also affect the trafficking of ATF6. However, the depletion of SNRPB results in a retardation of ER-to-Golgi transport, and not a complete block. Thus, the arrival of ATF6 to the Golgi might be slower, but it will nevertheless ultimately arrive and be cleaved there. Thus, we speculate that the peak of ATF6 activation might be delayed in SNRPB depleted cells. However, any regulator of ER-export would probably have the same effect.

11. To better understand the role of UPR machinery and SNRPB, additional experiments checking the downstream and upstream events to ATF6 function, such as ERp18 association and ATF6f induction targets are strongly recommended.

Response: We followed the reviewer's suggestion and investigated the role ERp18.

We first tested whether ERp18 knockdown affects the activation of ATF6. This is not a trivial task for us, because we could never detect ATF6 activation using immunoblotting (despite having tried extensively with multiple antibodies). We therefore used immunofluorescence against endogenous ATF6, which we found to localize in the ER and the Golgi. Treatment with thapsigargin resulted in a marked shift to the Golgi, which we used as a surrogate for the activation of ATF6. Silencing ERp18 resulted in a reduction in thapsigargin-induced Golgi accumulation. This indicates that silencing of ERp18 alters ATF6 activation, which is in line with the previous work by the Bulleid group (PMID: 28292422).

The dynamic range of this assay is not huge, but it allowed us to verify that ERp18 is indeed an ATF6 regulator in our hands. Having validated the effect of ERp18, we then tested whether the induction of SNRPB levels in GAT1 overexpressing cells is affected by ERp18 knockdown. This was the case, as shown in the new supplementary Figure EV9.

As for the downstream targets of ATF6, we are not why this would improve the current work. The focus of this work is not on the regulation of ATF6 or its downstream targets.

Minor points:

1. In figure 1D it would be interesting to also show a membrane marker to better identify the borders of the cells. This would provide a stronger point regarding the localization of collagen inside or outside the cells.

Response: unfortunately, due to technical difficulties, we could not do this. We tried using E-cadherin, which gave a strange pattern that labelled cell borders and their interior, indicating that the staining might be unspecific in our hands.

2. There are typos in the methods section --Immunofluorence-- staining and confocal microscopy: For ERES experiments, images were taken, focusing at nuclear ring and adjust the threshold level, when only puncta - signal-- is detected.

Response: the typos were removed

Referee #2:

Zahoor et al provide interesting evidence of a role for splicing in ER proteostasis. A combination of cell and mouse-based analysis point to the levels of the Sm protein SNRPB, and other Sm proteins, being important for ER-to Golgi trafficking and levels of SNRPB being important for bone development in mice making the link to the genetic disorder CCMS. On the whole the data is convincing, well presented, and provides an interesting mechanistic explanation for the developmental defects associated with CCMS. However, there are issues with the quality of some of the data and the manuscript contains many mistakes and missing information that requires correcting. In addition, the narrative implies that it is SNRPB on its own that regulates splicing where in fact the depletion of SNRPB almost certainly affects snRNP/spliceosome function or abundance to influence splicing which is not discussed by the authors.

Response: we thank the reviewer for the constructive criticism. As you will see from our responses below, we have taken the points of the reviewer seriously and changed the presentation such that we do not imply that this is an effect specific to SNRPB, but rather to the Sm ring components of the spliceosome. In fact, our original manuscript did show that other components (e.g. SNRPG or SNRPD1) have an effect on ER-export, while unrelated components such as SNRPB2 or SNRPC have no effect. We present new data to emphasize this and also write the text such that we do not make the impression that we think that this is a function carried out "privately" by SNRPB.

The manuscript starts with analysis of ER-to-Golgi trafficking using the RUSH assay with Mann and Col-X

reporters following knockdown of SNRNPB with siRNAs. These data convincingly show that SNRNPB levels are important for ER-to-Golgi trafficking. To determine if the ER-to-Golgi trafficking of a native cargo is influenced, the localisation of ERGIC-53 was analysed following SNRNPB knockdown. Unfortunately, the images and quantitation are not convincing/clear. An explanation/details of how peripheral ERGIC-53 was counted is needed somewhere. Could the images in Figure 1C be annotated to show the region of the cell considered the periphery?

Response: We include a screenshot of the analysis. First the image is thresholded such that ERGIC-53 positive structures are included. Then, we count only structures that are smaller than 10 pixel (to exclude the central region of ERGIC-53), but bigger than 0.5 pixels (to avoid counting noise) but smaller than 10 pixels (to avoid including larger clusters). In this particular cell, we counted 144 puncta in the SNRNPB depleted cell compared to 221 in the control cell.

Neural crest tissue from embryos with specific heterozygous neural crest cell knockout of SNRNPB were next investigated for localisation of collagen-I. Unfortunately, the change in collagen-I localization implied in Figure 1D is not convincing. It is not clear how one can tell between intracellular and extracellular localisation from the presented images. Can another intracellular marker be used to show increased intracellular co-localisation of collagen-I? In contrast, the RUSH assay for collagen-I in Figure 1E reveals that SNRNPB knockdown does influence collagen-I ER-to-Golgi trafficking.

Response:

Determining subcellular localization of collagen-I in tissue is not a trivial task. The resolution of such images is never sufficiently strong compared to cell culture data. Therefore, we are unfortunately not able to provide any convincing data to address the point of the reviewer directly. A biochemical fractionation from tissue samples is technically challenging, because we would need to extract extracellular collagen without damaging the cells in the tissue and then compare it with intracellular collagen. Performing such a fractionation without any cross-contamination between the intra- and extra-cellular fractions is unlikely to succeed.

We are aware that this is problematic and wanted to provide the reviewer with stronger evidence that there is indeed a trafficking defect for collagen. The data we had in the original manuscript only contained a RUSH assay with collagen trafficking in HeLa cells. Thus, we developed a trafficking assay for endogenous collagen-I in primary human fibroblasts. We cultured these cells in the absence of ascorbic acid and performed siRNA knockdown of SNRNPB. To initiate collagen-I trafficking, we treated cells with ascorbic acid and fixed them before and 40 min after treatment. We found that silencing SNRNPB resulted in a retardation of collagen-I trafficking from the ER to the Golgi (new Figure 1D). This result with endogenous collagen-I supports the notion of a trafficking defect for collagen in SNRNPB depleted cells.

We hope that these new data convince the reviewer that the trafficking defect of collagen-I is relevant and not just an observation restricted to artificial systems like the RUSH assay. We moved the tissue staining data to the supplement. In case the reviewer does not like our tissue data, we are ready to completely remove them from the manuscript. However, we think that we provide strong evidence that SNRNPB (and other components of the spliceosome) affect ER-to-Golgi trafficking in general, and of collagen-I even at the endogenous level.

We are not sure why the reviewer wrote "in contrast". It would imply that a stronger intracellular retention does not agree with a retardation of trafficking. To our understanding, intracellular localization and trafficking retardation are well in agreement. Maybe the reviewer meant that a RUSH assay with overexpressed collagen does not support this conclusion, because the collagen is overexpressed in contrast to endogenous collagen in mice. To address this point, we obtained primary human fibroblasts.

To further analyze the influence of SNRNPB knockdown on the number of ER exit sites (ERES), an important node in the proteostasis network, Sec31 was used as a marker. Figure 2A shows nicely that knockdown of SNRNPB in

HeLa cells decreases the number of ERES. The reader is next referred to Figure S1D for U2OS data but unfortunately this Figure was not included in the Supplementary materials.

Response: The figure that we were referring to was actually figure is S1C (Figure EV1C). We are sorry for this mistake.

In a potentially important experiment, fibroblasts from a CCMS patient were analysed for ERES numbers with and without rescue using a transfected GFP-SNRPB construct. While the quantitation presented comparing ERES numbers looks good, in the images presented in Figure 2B the Sec31 appears to be colocalising with SNRPB in the nucleus? All other non-electroporated cells have a very clearly defined nucleus without much Sec31 localization. Therefore, these images are not convincing. In addition, the information provided on the variant that the CCMS patient cells carry, 165G>C, is not informative. What type of variant is this? c.165G>C [p.Arg55Ser] would provide more information to the reader. Also, no information is given in the Methods on the source of patient fibroblasts or methods for transfection.

Response: ERES do not exist in the nucleus. ERES are subdomains of the ER and are not detached from this organelle as has been shown using EM-tomography (Zeuschner et al, Nat Cell Biol, 2005) and FIB-SEM experiments (Weigel et al, Cell, 2021). Thus, ERES do not localize to the nucleus. The appearance of ERES in the nuclear area is because fibroblasts are very flat cells and a confocal slice of around 0,75 µm would make ERES that are in the focal plane below the nucleus appear to lie within it.

The Method of transfection is now explained in the methods section. We used the Neon NxT electroporation system to transfect fibroblasts. The source of the fibroblasts is from one of the co-authors (Cormier-Daire), who had described this patient previously (Bacrot et al, Hum Mutat, 2015).

Now that it was established that SNRPB was required for ER-to-Golgi trafficking the authors investigated whether other Sm ring proteins were required for ER-to-Golgi trafficking. Knockdown of SNRPB, SNRPC, SNRPD1 and SNRPG was shown not to affect the expression of each other significantly. Using the RUSH assay again to monitor ER-to-Golgi trafficking revealed that only the Sm ring proteins SNRPB, SNRPD1 and SNRPG decreased ER-to-Golgi trafficking. This result is very interesting and indicates that general disruption of snRNPs/spliceosomes, and not just SNRPB alone, disrupts ER-to-Golgi trafficking (see comments below).

Response: We changed the wording in the abstract and the text to make this point clear.

The next important step in this work was to determine how reducing SNRPB levels specifically disrupted ER-to-Golgi trafficking. Levels of proteins involved in ER-to-Golgi trafficking such as Sec16A, Sar1A, Sec12, Sec24C, and Sec31 were investigated by western blotting. While there are potentially some noticeable reductions of some of these ERES proteins, these western blots (Figure 4A and EV4) would really need to be repeated, quantitated and statistical analysis carried out to bolster the significance of these results. It would also be important to see here whether knockdown of other Sm ring proteins influenced ERES protein levels. There appeared to be no influence of SNRPB knockdown on the mRNA levels of the ERES proteins.

Response: We performed the densitometry of these blots and show the result in the figure. We are not sure which blots to repeat, because the reduction in Sec16A, Sec24C, Sec12, Sar1A, and Sec31A is very clear. All other proteins are barely or not affected at all. This is supported now in the densitometry result. The densitometry was performed using Fiji.

We tested whether splicing of Sec16A is affected by other Sm-ring components and found that to be the case. These data show that SNRPB as well as other components of the Sm ring regulate Sec16A. On the other hand, proteins that are not part of the Sm-ring such as SNRPB2 as well as SNRPC have no effect on Sec16A splicing or on the levels of ERES proteins.

Now, as predicted, the knockdown of SNRPB will most certainly impact the splicing of certain pre-mRNAs and most likely pre-mRNAs of ERES components that have been shown to be reduced and involved in ER-to-Golgi trafficking. Concentrating on Sec16A, a clear defect in Sec16A pre-mRNA splicing and nuclear accumulation of unspliced pre-mRNA following SNRPB knockdown was observed, whereas two other pre-mRNAs were not affected. This result is interesting and nicely supports the idea that there is a splicing defect responsible for the defects observed following SNRPB knockdown. Again, it would be important to see here whether knockdown of other Sm ring proteins influenced Sec16A splicing.

Response: These data are now provided as figure 5A and supplementary Figure EV5.

We repeated the splicing experiments with SNRPB knockdown in response to a point reviewer-3. To address the concern of this reviewer about the specificity of the effect for sm-Ring components, we tested the effect knockdown of SNRPC (not part of the Sm-Ring) and found it to have almost no effect on Sec16A splicing compared to SNRPB. On the other hand, silencing SNRPG or SNRPD1 had an effect on Sec16A splicing.

To determine how important the mis-splicing of Sec16A is to ER-to-Golgi trafficking a RUSH assay was used to

determine the influence of providing exogenous Sec16A to rescue Sec16A function on ER-to-Golgi trafficking. Interesting a significant amount of ER-to-Golgi trafficking was restored by adding back Sec16A indicating that Sec16A plays a major role in ER export and is most likely one of the more important pre-mRNAs that is mis-spliced.

Global analysis of expression and splicing by RNA-seq of cells depleted of SNRPB was the logical next step. A volcano plot is presented to display differential expression and it is stated that there were not any expression changes in Sec16A or other components of the secretory pathway. It would have been appropriate here to also provide these differential expression data in a Supplementary Excel file to allow readers to see and interrogate the data themselves.

Response: We thought we had submitted these data as supplementary data, but something might have gone wrong during the submission process. We apologize for that and include these data now as a supplementary table.

Splicing analysis was carried out using two different bioinformatics tools. A typical pattern of splicing defects was observed as seen previously for depletion of other core splicing factor genes which is a change in skipped exons. Sec16A splicing was picked out to show its mis-splicing but the sashimi plot for the siSNRPB appeared not to be formatted correctly. It would have been nice here to have picked out one or two other significant/relevant mis-splices to confirm by RT-PCR and present as sashimi plots.

Response: We already confirmed the effect on the protein levels (by western blot) of GOSR1 supplementary Figure EV4. In addition, we show the sashimi plots for GOSR1 in this figure (EV4B).

Finally, as expected for this type of analysis, common features of pre-mRNA that displayed splicing changes were determined. Disappointingly, there is no indication that these RNA-Seq data will be made publicly available through any of the normal routes like GEO.

Response: the data have now been added as requested. The data have been deposited on the Gene Expression Omnibus (GEO) and are accessible under accession number GSE254122 (Reviewer Access Token: slmbckmmnbgdyh). We apologize for the oversight. It was not our intention to keep these data.

Next the hypothesis that SNRPB was either a mediator or effector of UPR was determined. ER overloading was used to determine the influence on SNRPB at both the mRNA and protein levels. An interesting upregulation of SNRPB is observed at the mRNA and protein levels which is determined to be ATF6 regulated. The western blots here for SNRPB levels should be repeated, quantitated and statistical analysis carried out to bolster the significance of these results. It is also important here to determine if the levels of other Sm ring protein mRNA and protein are increased following ER overload.

Response: densitometry was performed and is shown as supplementary Figure EV9.

An important part of this work is to link the functional work here on SNRPB and ER-to-Golgi trafficking to CCMS and bone development defects seen with CCMS. The authors tested whether loss of SNRPB affects osteogenesis using an in vitro assay where mesenchymal cells were differentiated to osteoblasts. Data is clearly presented indicating that knockdown of SNRPB does in fact inhibit in vitro osteogenesis and can be rescued by Sec16A overexpression. As before it would be important to at least include knockdown of another Sm ring protein to determine if effects are specific for SNRPB.

Response: We appreciate the point of the reviewer, and agree that it is a very good one. However, our circumstances do not allow us to address this experiment and we have discussed this matter with the Editor. We would like to also explain it to this reviewer. Firstly, the postdoc (1st author) of this paper is no longer part of the lab. This is because the Farhan group moved from Norway to Austria and the 1st author decided to remain in Norway. Thus, the technical expertise for this very demanding experiment is not available. The other very important factor is that the source of these cells was a neighbor lab in Norway and this lab closed due to the retirement of the PI. We hope for the understanding of the reviewer that these challenges make it very hard for us to address this point. We have also discussed this point with the Editor and explained the circumstances.

The authors conclude by presenting very nice results in mice with induced deletion of *Snrpb* in E8.5 mice and looking 6 days later at bone development in the appendages. The data convincingly indicate *Snrpb* is required for growth and patterning of the bones of the mouse appendages further indicating the link between SNRPB and the defects associated with CCMS.

The way the manuscript is written the authors are implying that SNRPB alone is acting as a "novel node" and is regulating ER proteostasis. As SNRPB is an Sm ring component, and the authors have also provided evidence that knockdown of other Sm ring components inhibits ER-to-Golgi trafficking, the most likely hypothesis is that reduced levels of SNRPB and other Sm ring components causes either defective snRNPs/spliceosomes or

reduced levels of functional snRNPs/spliceosomes to influence the splicing of a subset of pre-mRNAs required for regulating ER proteostasis. So the functional defect here is not caused by SNRPB alone but the effects of having less SNRPB have on the spliceosome. So the text should be revised throughout the manuscript to indicate that it is not SNRPB alone that is influencing ER proteostasis, but the effects of reduced SNRPB on the snRNPs/spliceosome. This point should also be emphasized in the Discussion.

Response: Reviewer is absolutely right and we should **change the wording and the title.**

Minor comments

Add to Abstract that knockdown of other Sm proteins inhibits ER-to-Golgi trafficking.

Response: done

At the beginning of the third paragraph of the Introduction the first two sentences repeat the same information: "To address knowledge gap, we focused on the spliceosomal subunit SNRPB for two main reasons. To investigate this further, we focused on the spliceosomal subunit SNRPB for two reasons."

Response: the mistake was fixed

Please define ERGIC in the first paragraph of the Results.

Response: done

In the third paragraph of the Results, in the first sentence, it would be useful to state here that you are using Sec31 localization as an ERES marker.

Response: done

Figure 1D - please define in Figure 1 legend what the annotations in the figure "NECT" and "HM" represent.

Response: NECT= neuroectoderm and HM= head mesenchyme. We added this info to the figures legend (new supplementary Figure EV2).

Figure 2B legend - quantification of what? Define what Pos. and Neg. means in legend.

Response: done

Figure S1 legend, first line - "siTNA" should be "siRNA"

Response: done

Figure 6D legend - please indicate whether this volcano plot was produced from data derived from RMATS, Whippet or both.

Response: it is based on RMATS, which is specified in the figure legend now.

Figure 8A - please provide some information in the legend as to how the quantification at the bottom of the panel was carried out.

Response: done

The following sentences are not worded clearly. "We plotted the changed exons in a volcano plot in Figure 6D, again highlighting exons in ER-Golgi trafficking genes in green. Overall, we identified 40 genes that exhibited affected transcripts." Was this 40 genes overall or 40 ER-Golgi trafficking genes?

Response: these 40 genes are from ER and Golgi only.

It is not clear what the siRNAs for SNRPB are. The methods say that siRNAs are listed in Supplementary Table S2 but I could not find that table. The only table I did find with siRNA information was in the "Key resources used" section which listed "siGenome Smartpool_SNRPB" and Silencer Select siRNA_HS_SNRPB". It should be added to this table which of these siRNAs are the siSNRPB#1 and siSNRPB#2 that are described in the presented data.

Response: apparently something went wrong with the submission of the tables. We hope that the table is now included and includes all necessary information.

Overall writing style issues. Use of unqualified "This" at the start and within many sentences. Please specifically define what "This" is.

Referee #3:

In this interesting and elegant manuscript, the authors demonstrate that proteins involved in the splicing machinery are also involved in the regulation of ER proteostasis and provide a new model and mechanism in

support of this. The data appear quite solid while the manuscript reads very well and the conclusions are balanced and supported by the experimental evidence. I only have a few minor comments:

1. In the second paragraph of the Introduction, there is a repeated sentence: "To address knowledge gap, we focused on the spliceosomal subunit SNRPB for two main reasons." should be deleted.

Response: the mistake was fixed

2. In the first section of the results, there is a reference to Figure S1D but this does not exist. I think it should say Figure 1C. Moreover, it is unclear why supplementary figures are labeled as EV1, EV2, EV3, etc....

Response: We apologize for the mistake. S1D should be S1C. This was fixed. The designation of the figures as EV1,2,3,etc..... was made as per the style of EMBOJ to refer to the supplementary figures as expanded view figures (hence EV). We realize that our labeling of the figures in the text as S1,2,etc.. might have been confusing and changed this now in the revised version.

3. It is a bit unclear why in Figure 5A there appears to be multiple bands in the Sec16A without the siSNRPB.

The authors describe this as potential splicing intermediates, but why they don't show in the other PCR reactions (for CANX and ACTB)? I guess it has to do with the primer annealing locations for each of those targets.... This is just a minor comment and the author may address it if they wish.

Response: These experiments have now been repeated with all appropriate control. We also included other Sm-ring components as well as a protein that is not part of the Sm-ring. We hope that this addresses the point of the reviewer. The new data are shown in Figure 5 and supplementary Figure EV5.

4. In Figure 9AB the labeling is barely readable and I would suggest to increase the fonts of those abbreviations.

5. In the last section of the results, the word 'pelvic' should read pelvis ('In the pelvic, the iliac bone was thinner....')

Response: the mistake was fixed.

Dear Dr Farhan,

Thank you for submitting a revised version of your manuscript. Your study has now been seen by all original referees. Referees #1 and #3 find that their previous concerns have been addressed and now recommend publication of the manuscript. Unfortunately, referee #2 finds that several concerns from the original report were not sufficiently addressed and does not recommend publication of the manuscript in its current form. These concerns include the lack of replication experiments and their quantification, a request to include revisions and a clearer description of these experiments in the manuscript, clearer labeling of experimental data in the figures/tables and improving the language throughout the manuscript. While this list seems to be extensive, I feel that these requests are reasonable, and addressable and I would therefore invite you to revise the manuscript addressing all the concerns raised by referee #2.

When preparing your letter of response to the referees' comments, please bear in mind that this will form part of the Review Process File and will therefore be available online to the community. For more details on our Transparent Editorial Process, please visit our website: <https://www.embopress.org/page/journal/14602075/authorguide#transparentprocess>

Please feel free to contact me if you have any further questions regarding the revision. Thank you for the opportunity to consider your work for publication, and I look forward to your revision.

With best regards,

Cornelius Schneider

Cornelius Schneider, PhD
Editor
The EMBO Journal
c.schneider@embojournal.org

We realize that it is difficult to revise to a specific deadline. In the interest of protecting the conceptual advance provided by the work, we recommend a revision within 3 months (29th Jul 2024). Please discuss the revision progress ahead of this time with the editor if you require more time to complete the revisions.

Referee #1:

The authors performed multiple experiments to address and elucidate the issues raised. The results are convincing and elevate the message of the paper.

Furthermore, the authors were thorough and showed great effort in their response to the comments, providing additional data to support their findings.

As a response to their question regarding major point 7, I appreciate the detailed answer and points raised and agree with the omission of the luciferase activity graph.

The new results and data presented strengthen the paper conclusions, quality and impact to the field, making it suitable for publication in The EMBO Journal.

Referee #2:

Below are the points that were not sufficiently addressed, or not addressed at all, in the responses to referees and the revised manuscript.

Author Response: we thank the reviewer for the constructive criticism. As you will see from our responses below, we have taken the points of the reviewer seriously and changed the presentation such that we do not imply that this is an effect specific to SNRPB, but rather to the Sm ring components of the spliceosome. In fact, our original manuscript did show that other components (e.g. SNRPG or SNRPD1) have an effect on ER-export, while unrelated components such as SNRPB2 or SNRPC have no effect. We present new data to emphasize this and also write the text such that we do not make the impression that we think that this is a function carried out "privately" by SNRPB.

Comments: Yes, in the Abstract, Introduction and Results the narrative has been changed to emphasize that effects are not specific for SNRPB but this narrative is missing from the Discussion. Please see other comments below.

Author Response: We include a screenshot of the analysis. First the image is thresholded such that ERGIC-53 positive structures are included. Then, we count only structures that are smaller than 10 pixel (to exclude the central region of ERGIC-53), but bigger than 0.5 pixels (to avoid counting noise) but smaller than 10 pixels (to avoid including larger clusters). In this particular cell, we counted 144 puncta in the SNRPB depleted cell compared to 221 in the control cell.

Comment: Thank you for this information, it is now clear how this analysis was undertaken. However, why was this information not provide in the revised manuscript? These images should be included in Figure 1C and the description of the analysis added to the Materials and Methods.

Author Response: Determining subcellular localization of collagen-I in tissue is not a trivial task. The resolution of such images is never sufficiently strong compared to cell culture data. Therefore, we are unfortunately not able to provide any convincing data to address the point of the reviewer directly. A biochemical fractionation from tissue samples is technically challenging, because we would need to extract extracellular collagen without damaging the cells in the tissue and then compare it with intracellular collagen. Performing such a fractionation without any cross contamination between the intra- and extra-cellular fractions is unlikely to succeed. We are aware that this is problematic and wanted to provide the reviewer with stronger evidence that there is indeed a trafficking defect for collagen. The data we had in the original manuscript only contained a RUSH assay with collagen trafficking in HeLa cells. Thus, we developed a trafficking assay for endogenous collagen-I in primary human fibroblasts. We cultured these cells in the absence of ascorbic acid and performed siRNA knockdown of SNRPB. To initiate collagen-I trafficking, we treated cells with ascorbic acid and fixed them before and 40 min after treatment. We found that silencing SNRPB resulted in a retardation of collagen-I trafficking from the ER to the Golgi (new Figure 1D). This result with endogenous collagen-I supports the notion of a trafficking defect for collagen in SNRPB depleted cells. We hope that these new data convince the reviewer that the trafficking defect of collagen-I is relevant and not just an observation restricted to artificial systems like the RUSH assay. We moved the tissue staining data to the supplement. In case the reviewer does not like our tissue data, we are ready to completely remove them from the manuscript. However, we think that we provide strong evidence

that SNRPB (and other components of the spliceosome) affect ER-to-Golgi trafficking in general, and of collagen-I even at the endogenous level.

Comments: These data are now improved and convincing with the addition of the primary human fibroblast results.

Author Response: ERES do not exist in the nucleus. ERES are subdomains of the ER and are not detached from this organelle as has been shown using EM-tomography (Zeuschner et al, Nat Cell Biol, 2005) and FIB-SEM experiments (Weigel et al, Cell, 2021). Thus, ERES do not localize to the nucleus. The appearance of ERES in the nuclear area is because fibroblasts are very flat cells and a confocal slice of around 0,75 µm would make ERES that are in the focal plane below the nucleus appear to lie within it. The Method of transfection is now explained in the methods section. We used the Neon NxT electroporation system to transfect fibroblasts. The source of the fibroblasts is from one of the co-authors (Cormier-Daire), who had described this patient previously (Bacrot et al, Hum Mutat, 2015).

Comments: Yes, I completely agree the ERES does not exist in the nucleus, that was the point I was trying to make as the data in Figure 2B makes it appear that Sec31 localized to the nucleus in the cell transfected with GFP-SNRPB. If you are saying that because the fibroblasts are so flat that Sec31 appears to lie within the nucleus then please provide this information to the reader in the text when you are describing Figure 2B. Thank you for providing the source of the fibroblasts, can you please include this information in the Materials and Methods of the manuscript?

Author Response: We performed the densitometry of these blots and show the result in the figure. We are not sure which blots to repeat, because the reduction in Sec16A, Sec24C, Sec12, Sar1A, and Sec31A is very clear. All other proteins are barely or not affected at all. This is supported now in the densitometry result. The densitometry was performed using Fiji. We tested whether splicing of Sec16A is affected by other Sm-ring components and found that to be the case. These data show that SNRPB as well as other components of the Sm ring regulate Sec16A. On the other hand, proteins that are not part of the Sm-ring such as SNRPB2 as well as SNRPC have no effect on Sec16A splicing or on the levels of ERES proteins.

Comments: I asked for the blots in Figure 4A and EV4 to be repeat, quantitated and statistical analysis carried out for these western blotting data. While Figure 4A was repeated and quantitated, no statistical analysis was carried out. Figure EV4 (now EV7) was not repeated, quantitated or had any statistical analysis carried out.

Author Response: These data are now provided as figure 5A and supplementary Figure EV5. We repeated the splicing experiments with SNRPB knockdown in response to a point reviewer-3. To address the concern of this reviewer about the specificity of the effect for sm-Ring components, we tested the effect knockdown of SNRPC (not part of the Sm-Ring) and found it to have almost no effect on Sec16A splicing compared to SNRPB. On the other hand, silencing SNRPG or SNRPD1 had an effect on Sec16A splicing.

Comments: New RT-PCR analysis of SEC16A splicing has been included Figure 5A and Figure EV5 after knockdown of SNRPB, SNRPC, SNRPG and SNRPD1 which would potentially supply interesting information of the specificity of Sm protein knockdown on SEC16A splicing. Unfortunately, these new data are not convincing, hard for the reader to interpret and incomplete. In the RT-PCR images (both Figure 5A and EV5) it is hard to see the differences in the unspliced products between the different siRNA treatments. In addition, these images should be annotated with arrows to point out the unspliced product of interest and quantitation of the unspliced product provide to show more clearly any increases in the unspliced product. Also, why was SNRPD1 knockdown not included in Figure 5A analysis of exon 6-8 splicing and SNRPG knockdown not included in Figure EV5 for analysis of exon 17-19 splicing?

Author Response: We thought we had submitted these data as supplementary data, but something might have gone wrong during the submission process. We apologize for that and include these data now as a supplementary table.

Comments: Thank you for now providing these RNA-seq data but what is the Supplementary Table number you have added? I assume you are referring to Table EV1? If Table EV1 contains the analysis of the expression data the table would be much more informative if it was given a title and some information/legend to provide the reader with information on what all the numbers/values represent.

Author Response: We already confirmed the effect on the protein levels (by western blot) of GOSR1 supplementary Figure EV4. In addition, we show the sashimi plots for GOSR1 in this figure (EV4B).

Comments: Confirmation of RNA-seq splicing data by RT-PCR for a few examples of mis-splicing is a routine and very straight forward experiment to be sure the RNA-Seq data is correct and provides a robust control for the RNA-Seq data. It is disappointing this analysis was not carried out. I assume here you are referring to Supplementary Figure EV7 and not EV4?

Previous comments not addressed:

1) The text should be revised throughout the manuscript to indicate that it is not SNRPB alone that is influencing ER proteostasis, but the effects of reduced SNRPB on the snRNPs/spliceosome. This point should also be emphasized in the

Discussion.

Response: Reviewer is absolutely right and we should change the wording and the title.

Comments: The authors have not addressed in the Discussion the point that reduced SNRPB or other Sm proteins will be influencing the snRNPs/spliceosome. As it stands the Discussion is still minimal/incomplete and does not fully discuss the implications of the work in sufficient detail, especially around the effects of reduce SNRPB or other Sm protein on the snRNPs/spliceosome.

2) This referee thinks that the manuscript would benefit from a more clear and precise writing style. In particular the use of unqualified "This" at the start and within many sentences. Please specifically define what "This" is.

For example, in the Abstract there is the sentence "This indicates that the role of SNRPB in osteogenesis is linked to its effects on ER export." What is "this"? If you don't label your pronoun, I could point it at any noun in the previous sentence (or even several sentences before). Always put a qualifier with the "this": "This **rescue** indicates that the role of SNRPB in osteogenesis is linked to its effects on ER export." You need to be specific and precise in scientific writing. Using an unqualified "this" to start a sentence can lead to a reader making their own, possibly incorrect, conclusions about what you are trying to say.

There are a number of places where the authors still use the phrases like "To support this notion..", "To this end..", "To this conjecture..", "This might be indicative of..", "This is in line with..", "This would be in line with.." "To link this effect..", "This is supported by.." "Further support for this notion.."

Also, at the beginning of the Discussion there is the sentence "This was based on the assumption that the splicing machinery is not a rate-limiting factor in this process." which provides the reader with absolutely no information as it contains an unqualified "this" in two places.

Please revise the manuscript to define what "this" is in all these cases. Please say what you are exactly referring to in the previous sentences in all these cases?

Minor comment

Figure 6 legend - do you mean volcano "plot" not volcano "blot"

Referee #3:

The authors have properly addressed all of my original comments and also performed additional experiments requested by the other reviewers which make this manuscript solid and not over-interpreted. I commend the authors for the clarity in their responses to the reviewers.

I have no further comments.

Below are our responses to the remaining points raised by Reviewer-2:

1. Comments: Yes, in the Abstract, Introduction and Results the narrative has been changed to emphasize that effects are not specific for SNRPB but this narrative is missing from the Discussion. Please see other comments below.

Response: We dealt with all these issues in the Discussion and hope that it is now satisfactory

2. Comment: Thank you for this information, it is now clear how this analysis was undertaken. However, why was this information not provide in the revised manuscript? These images should be included in Figure 1C and the description of the analysis added to the Materials and Methods.

Response: The description of the method was added to the Methods section. After consultation with the Editor, we included the explanatory figure as Supplementary Figure EV11, which is referred to in the Methods section. We thought that this figure is too technical in nature to be included into the main figures.

3. Comments: Yes, I completely agree the ERES does not exist in the nucleus, that was the point I was trying to make as the data in Figure 2B makes it appear that Sec31 localized to the nucleus in the cell transfected with GFP-SNRPB. If you are saying that because the fibroblasts are so flat that Sec31 appears to lie within the nucleus then please provide this information to the reader in the text when you are describing Figure 2B. Thank you for providing the source of the fibroblasts, can you please include this information in the Materials and Methods of the manuscript?

Response: We added the source of the fibroblasts to the Methods sections and we added the description of how ERES were counted as well.

4. Comments: I asked for the blots in Figure 4A and EV4 to be repeat, quantitated and statistical analysis carried out for these western blotting data. While Figure 4A was repeated and quantitated, no statistical analysis was carried out.

Response: Apologies for the oversight. Statistics were added.

5. Figure EV4 (now EV7) was not repeated, quantitated or had any statistical analysis carried out.

Response: In agreement with the Editor, we decided that this experiment would not benefit the entire manuscript, because it will not add anything towards the conclusion of the story.

6. Comments: New RT-PCR analysis of SEC16A splicing has been included Figure 5A and Figure EV5 after knockdown of SNRPB, SNRPC, SNRPG and SNRPD1 which would potentially supply interesting information of the specificity of Sm protein knockdown on SEC16A splicing. Unfortunately, these new data are not convincing, hard for the reader to interpret and incomplete. In the RT-PCR images (both Figure 5A and EV5) it is hard to see the differences in the unspliced products between the different siRNA treatments. In addition, these images should be annotated with arrows to point out the unspliced product of interest and quantitation of the unspliced product provide to show more clearly any increases in the unspliced product. Also, why was SNRPD1 knockdown not included in Figure 5A analysis of exon 6-8 splicing and SNRPG knockdown not included in Figure EV5 for analysis of exon 17-19 splicing?

Response: We added the arrows as indicated by the reviewer and restructured Figure 5A such that it is very easy to follow what is going on.

To harmonize between the Figures, we have no concentrated on doing the splicing experiments for Sec16A for SNRPB, SNRPC and SNRPD1. Figure 5 shows exons 6-8 and supplementary Figure EV5A shows exons 17-19. As for quantification, we are not sure what to quantify here. To our understanding, this type of experiment is not quantitative, but only semi-quantitative. We don't think it is right to quantify these types of data.

7. Comments: Thank you for now providing these RNA-seq data but what is the Supplementary Table number you have added? I assume you are referring to Table EV1? If Table EV1 contains the analysis of the expression data the table would be much more informative if it was given a title and some information/legend to provide the reader with information on what all the numbers/values represent.

Response: the explanation of what the table contains in provided in the file "Supplement". We hope that this is what the reviewer was referring to.

8. Comments: Confirmation of RNA-seq splicing data by RT-PCR for a few examples of mis-splicing is a routine and very straight forward experiment to be sure the RNA-Seq data is correct and provides a

robust control for the RNA-Seq data. It is disappointing this analysis was not carried out. I assume here you are referring to Supplementary Figure EV7 and not EV4?

Response: We decided to test the splicing of Exon 7 of GOSR1 which was found to be affected in our RNAseq data (supplementary table EV2). This is now shown in Figure EV5B.

In addition, we picked exon 20 of Sec31 and of Exon 2 of ERGIC2, where we also find evidence that there is a splicing defect in SNRPB depleted cells. These data are shown now in Figure EV5C. The figure is also labeled with arrows explaining the problem.

9. This referee thinks that the manuscript would benefit from a more clear and precise writing style. In particular the use of unqualified "This" at the start and within many sentences. Please specifically define what "This" is.

Response: all these issues were dealt with and the text was carefully checked. In the first revision, we were not sure what the reviewer meant. After the reviewer explained himself, we were able to address his/her comment and we changed the text as requested.

Dear Hesso,

Thank you for submitting a revised version of your manuscript. We find that you have addressed all the additional remarks raised by the referee. There remain only a few mainly editorial points that have to be addressed before I can extend formal acceptance of the manuscript:

1. FUNDING INFO: missing info in eJP: H2020 MSCA-innovative training networks SECRET and SAND; Research Council of Norway through its Centres of Excellence funding scheme (Project: 262652); Canadian Institutes of Health Research (CIHR), bridge funding from the Research Institute of McGill University Health Centre (RI-MUHC) and the Azrieli Foundation; Fonds de Recherche du Québec (FQRS); missing info in ms file: Austrian Science Fund (FWF) FG2000
2. Please add up to 5 keywords
3. Please add a DATA AVAILABILITY SECTION where you summarize all datasets used in the manuscript together with the links to the data depositories.
4. Please add the "DISCLOSURE AND COMPETING INTERESTS STATEMENT" section.
5. Please rename the supplementary figures and tables
DATASET EV LEGENDS: Supplementary Tables EV1-EV3 should be renamed to Dataset EV1-EV3 with the corresponding callouts; legends should be removed from "Expanded View Figures and legends" and uploaded in the respective Excel files as a separate tab
APPENDIX 1 FILE WITH ToC: "Expanded View Figures and legends" should be renamed to Appendix file, the nomenclature should be Appendix Figure S1-S10 with the appropriate callouts; on the title page there should be a ToC with the page numbers, and legends should be placed below the corresponding figures (other option is to reorganize the supplementary figures: we can have up to 5 "Expanded View" figures (=second level), whose legends would also need to be in the main text, and which would be type-set and directly visible (expandable) with the HTML version of the paper - these figure should be named and references as "Figure EV1-5" and uploaded as individual figure files in high-resolution. Any additional data figures (third level below main and EV figures) should be included in a single Appendix PDF, with their respective legends below each figure, and named/referenced as "Appendix Figure S1/2/3". The Appendix should furthermore be prefaced by a brief Table of Contents, listing the included figures and their respective page numbers.)
6. Please provide SD and compile the SD checklist
7. Synopsis:
Papers published in The EMBO Journal are accompanied online by a 'Synopsis' to enhance discoverability of the manuscript. It consists of A) a short (1-2 sentences) summary of the findings and their significance, B) 3-4 bullet points highlighting key results and C) a synopsis image that is 550x300-600 pixels large (width x height, jpeg or png format). You can either show a model or key data in the synopsis image. Please note that the image size is rather small and that text needs to be readable at the final size. Please send us this information together with the revised manuscript.
8. Figure 5C GAPDH looks to be overexposed. Please provide SD for this figure and a corrected figure at natural brightness.
9. For figures with multiple panels with similar experimental setups we recommend a separate 'Data Information' section in the figure legend which summarizes the experimental setup (the legends of figures 9a-e).
10. The figure title is missing for the legends of figures EV 1; 2.
11. Please define the annotated p values ***/**/* in the legend of figure 8a; EV 1a, c; as appropriate.
12. Please indicate the statistical test used for data analysis in the legends of figures 6a, d; EV 1a, c."
13. Please note that the box plots need to be defined in terms of minima, maxima, centre, bounds of box and whiskers, and percentile in the legends of figures 2b; EV 6; EV 7b; EV 8.
14. Please note that information related to n is missing in the legends of figures 1b, d; 2a; 4b; 5b; 7e-f; 8c; 9g; EV 1a, c; EV 4; EV 6; EV 8; EV 9a-b; EV 10a.
15. Although 'n' is provided, please describe the nature of entity for 'n' in the legend of figure EV 7b.
16. Please note that the error bars are not defined in the legends of figures 4a-b; 5b; 7e-f; 8a, c; Ev 1a; EV 4; EV 9a-b."
17. Please note that the scale bar needs to be defined for figures 1a-d; 2a; 7b; EV 2b; EV 3.
18. Please note that scale bar and its definition are missing for figures 2b; 3a; EV 2a."
19. Section order should be corrected: title page with complete author information, abstract, keywords, introduction, results, discussion, materials & methods, data availability section, acknowledgements, disclosure and competing interests statement, references, main figure legends, tables, expanded figure legends.

With best regards,

Cornelius

Cornelius Schneider, PhD
Editor | The EMBO Journal
c.schneider@embojournal.org

We realize that it is difficult to revise to a specific deadline. In the interest of protecting the conceptual advance provided by the work, we recommend a revision within 3 months (23rd Sep 2024). Please discuss the revision progress ahead of this time with the editor if you require more time to complete the revisions.

The authors have addressed all minor editorial requests.

Dear Prof. Farhan,

I am pleased to inform you that your manuscript has been accepted for publication in the EMBO Journal.

Yours sincerely,

Cornelius Schneider, PhD
Editor
The EMBO Journal
c.schneider@embojournal.org
